# The Double-Edged Nature of the Rashomon Set
# for Trustworthy Machine Learning

Ethan Hsu [* 1]  Harry Chen [* 2]  Chudi Zhong [3]  Lesia Semenova [4]

## Abstract

Real-world machine learning (ML) pipelines rarely produce a single model; instead, they produce a Rashomon set of many near-optimal ones. We show that this multiplicity reshapes key aspects of trustworthiness. At the individual-model level, sparse interpretable models tend to preserve privacy but are fragile to adversarial attacks. In contrast, the diversity within a large Rashomon set enables reactive robustness: even when an attack compromises one model, a practitioner can switch to a different near-optimal model that remains accurate, without retraining. However, the same diversity increases information leakage, as disclosing more near-optimal models provides an attacker with progressively richer views of the training data. This produces a robustness–privacy trade-off governed by diversity, which we analyze theoretically and empirically. Beyond this trade-off, Rashomon sets are stable under small distribution shifts, so a set computed once remains valid under such shifts without re-computation. Our results[1] highlight the dual role of Rashomon sets as both a resource and a risk for trustworthy ML.

## 1. Introduction

In high-stakes domains such as lending, criminal justice, and healthcare, the standard goal of finding a single "best" predictive model is no longer enough. A long-standing observation in statistical modeling challenges the idea that such a best model exists. Breiman's Rashomon Effect (Breiman,

---

[*]Equal contribution  [1]Duke University, Durham, NC, USA
[2]Massachusetts Institute of Technology, Cambridge, MA, USA
[3]University of North Carolina at Chapel Hill, Chapel Hill, NC, USA [4]Rutgers University, New Brunswick, NJ, USA. Correspondence to: Ethan Hsu <ethan.hsu.contact AT gmail.com>, Harry Chen <harryc60 AT mit.edu>.

*Proceedings of the 43rd International Conference on Machine Learning*, Seoul, South Korea. PMLR 306, 2026. Copyright 2026 by the author(s).

[1]https://github.com/EtHsu0/
rashomon-duality

2001) and follow-up work (Semenova et al., 2022; Marx et al., 2020; Black et al., 2022; Ganesh et al., 2025; Paes et al., 2023; Rudin et al., 2024) show that many distinct models can achieve nearly indistinguishable predictive performance while relying on different features, logic, or decision boundaries. This multiplicity matters in modern high-stakes settings, where institutions require models that are not only accurate but also interpretable, stable under distribution shifts, and privacy-preserving. Recent algorithms (Xin et al., 2022; Zhong et al., 2023; Liu et al., 2022; Hsu et al., 2024a;b; Donnelly et al., 2025; Feng et al., 2025) can construct or approximate these sets of good models, known as Rashomon sets, and study and use them in practice.

Crucially, modern machine learning pipelines routinely produce such multiplicity even when practitioners do not think of themselves as computing a Rashomon set. Hyperparameter sweeps, fairness constraints, random seeds, feature restrictions, and automated model search all generate many near-optimal models. These models may often be inspected, for example, in robustness audits or regulatory reporting. This perspective motivates a shift. Rather than viewing the Rashomon set as a theoretical construct, we argue that in realistic governance scenarios the *Rashomon set itself is the natural policy object*. It is the set of near-optimal models that institutions already generate during model development, that shape downstream decisions, and that regulators, auditors, or internal teams may query, compare, stress-test, or even partially disclose. Once the Rashomon set is treated as a policy object, a natural question arises: *What are the positive and negative trustworthiness consequences of having a large, diverse Rashomon set?*

Although interpretability and fairness have been extensively studied within Rashomon sets (Semenova et al., 2022; 2023; Coston et al., 2021; Laufer et al., 2025; Dai et al., 2025), the relationships between model multiplicity and robustness, privacy, and stability remain far less understood. In this work, we focus on these under-explored aspects of trustworthiness and examine how large, diverse Rashomon sets shape robustness, stability, and information leakage. Importantly, these sets introduce both opportunities and risks.

On the positive side, diversity within the Rashomon set can be a resource. When monitoring, audits, or red-teaming

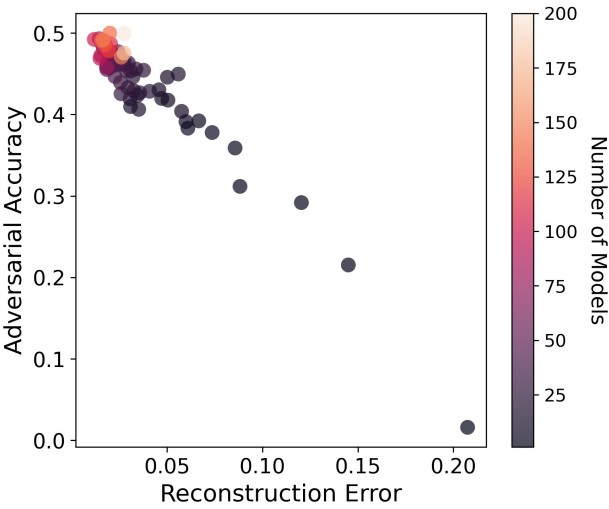

*Figure 1.* The robustness–privacy trade-off on the COMPAS dataset: with the increasing diversity in the Rashomon set, adversarial accuracy increases (greater reactive robustness), while reconstruction error decreases (greater information leakage).

*Table 1.* Comparison of trustworthiness criteria at the single-model level versus the Rashomon-set level.

| Criterion | Single Sparse Model | Rashomon Set |
|---|---|---|
| Robustness | ✗ vulnerable (no alternatives) | ✓ reactive robustness (has alternatives) |
| Stability | ✓ algorithmic stability (via regularization) | ✓ stable set (models remain near-optimal under shifts) |
| Privacy | ✓ private (sparse model) | ✗ leakier (more models reveal more information) |

The remainder of this paper explains *how this phenomenon arises* and discusses its implications.

We take a conceptual approach, supported by theoretical and empirical evidence, to study robustness, stability, and information-theoretic privacy in learning problems that admit many near-optimal models. By working with decision trees and linear models, we make the mechanisms underlying these properties explicit, while experiments with multilayer perceptrons demonstrate that trends extend to more complex architectures. These trustworthiness criteria behave qualitatively differently at the level of a single model than at the level of the Rashomon set (see Table 1), which we argue is the natural object of analysis in modern pipelines.

## 2. Related Work

**Rashomon Effect.** The Rashomon Effect (Breiman, 2001) refers to the existence of many equally accurate models, also called model multiplicity (Marx et al., 2020). Recent work develops ways to measure this effect (Semenova et al., 2022; Hsu & Calmon, 2022; Marx et al., 2020; Xin et al., 2022; Hsu et al., 2024b;a) or study it through the lens of simplicity (Semenova et al., 2023; Boner et al., 2024), fairness (Langlade et al., 2025; Dai et al., 2025; Meyer et al., 2025), explainability (Turbal et al., 2026; Müller et al., 2023), variable importance (Fisher et al., 2019; Donnelly et al., 2023; 2026), robustness (Nguyen et al., 2025), and differential privacy (Kulynych et al., 2023). While Black et al. (Black et al., 2022) focuses on the social and normative implications of model multiplicity, our work studies how a diverse Rashomon set reshapes trustworthiness properties, such as robust model selection and information leakage.

Hsu et al. (2026) provide empirical evidence that the Rashomon set of sparse decision trees contains individual models that perform as well as models obtained by directly optimizing for trustworthy criteria. We instead study the Rashomon set as a whole, showing theoretically and empirically that its multiplicity is double-edged: robustness and privacy are in fundamental tension, as the same diversity that enables reactive robustness amplifies privacy leakage.

**Adversarial Robustness.** Many ML models are sensitive to perturbations in the inputs that lead to changes in the model

reveal a problematic region of the input space, institutions need not retrain from scratch. Instead, they can select a different near-optimal model in the Rashomon set that behaves more favorably on the flagged inputs. Furthermore, the Rashomon set can be leveraged for moving target techniques (Amich & Eshete, 2021; He et al., 2025), where models can be rotated out either on a regular basis or in response to adversarial attacks. A diverse Rashomon set can significantly reduce the risk of attacks transferring between released models and other near-optimal models. We refer to this ability to switch to a differently behaved, near-optimal model as *reactive robustness*. It arises because the Rashomon set contains many models that are equally accurate yet rely on different features or decision boundaries, increasing the chance that at least one model avoids the vulnerability or failure mode discovered in deployment.

On the negative side, the same diversity can be a liability. Releasing or internally exposing many near-optimal models, even individually sparse and interpretable ones, can accelerate information leakage about the training data. Each additional model offers a new "view" of the dataset, tightening an attacker's ability to reconstruct features or approximate the underlying distribution.

An empirical illustration of this duality appears in Figure 1, which uses the COMPAS recidivism dataset. Each point represents a collection of sparse decision trees selected from the Rashomon set, varying both how many and which trees are included. The trend reveals a *robustness–privacy trade-off induced by model multiplicity*: diversity provides robustness but simultaneously increases information leakage. We observe similar behavior across multiple datasets (Section 6).

outputs (Nguyen et al., 2015; Finlayson et al., 2019; Malik et al., 2024). Prior work analyzes and mitigates this vulnerability by injecting random noise into inputs (Li et al., 2019; Guo et al., 2017; Delattre et al., 2025) or model weights (Pinot et al., 2019; He et al., 2019; Wang et al., 2025), regularizing for robustness (Goodfellow et al., 2015; Zhang et al., 2019), removing (Liao et al., 2018; Liu et al., 2025) or detecting (Metzen et al., 2017) the adversarial noise within the inputs. Theoretically, prior work decomposes adversarial error into natural and boundary components tied to the distance to the decision boundary (Zhang et al., 2019; Li & Yuanzhi, 2026), and shows that wider, more complex networks tend to be more vulnerable (Wu et al., 2021). Yao et al. (2025) show that ensemble diversity among surrogate models strengthens adversarial attacks by improving transferability. We study the dual side of this phenomenon from the defender's perspective: when diverse models form a Rashomon set, the same disagreement across near-optimal models helps to block such transferability (Section 5.1).

Stability and its connection to generalization have also been studied (Bousquet & Elisseeff, 2002; Feldman & Vondrak, 2018). Notably, adversarial perturbations often transfer across models (Hosseini et al., 2017; Szegedy et al., 2013; Tao et al., 2026).

**Privacy and information leakage.** Privacy in ML concerns preventing models from leaking sensitive training information (Liu et al., 2021). Classic mechanisms include k-anonymity (Sweeney, 2002), $\ell$-diversity (Machanavajjhala et al., 2007), and differential privacy (Dwork, 2006), while empirical studies rely on membership-inference and inversion attacks (Shokri et al., 2017; Fredrikson et al., 2015; Song & Mittal, 2021; Noorbakhsh et al., 2024; Hu et al., 2025). Information-theoretic formulations measure leakage via mutual information between the learned model and data (Bassily et al., 2018; Wang et al., 2021), and prior work connects these notions to differential privacy (Alvim et al., 2011b;a; Rassouli & Gündüz, 2019). We instead show that individual sparse models are private, but collections of many models can leak far more information.

**Robustness-Privacy trade-off.** Trade-offs among accuracy, robustness, and privacy are well studied in trustworthy ML (Gittens et al., 2022; Allouah et al., 2023), particularly accuracy–privacy and accuracy–robustness trade-offs (Bassily et al., 2014; Wang et al., 2017; Shafahi et al., 2018; Tsipras et al., 2018; Zhang et al., 2019; Nhan et al., 2025). The robustness–privacy relationship remains open: some work suggests they can reinforce each other (Dwork & Lei, 2009; Lecuyer et al., 2018; Phan et al., 2020), while others find differential privacy can reduce robustness (Tursynbek et al., 2020; Boenisch et al., 2021; Wu et al., 2023) and adversarial training can increase privacy risk (Song et al., 2019; He et al., 2020; Zhang et al., 2025). However, prior work

typically analyzes these trade-offs at the level of individual models or training procedures, rather than at the Rashomon set level.

## 3. Notations

Consider $n$ i.i.d. samples $S = \{(x_i, y_i)\}$ from an unknown distribution $\mathcal{D}$ on $\mathcal{X} \times \mathcal{Y}$, where $\mathcal{X} \subset \mathbb{R}^p$ and $\mathcal{Y} \subset \mathbb{R}$ are the input and output spaces respectively. Let $\mathcal{F}$ be a hypothesis space. We focus on interpretable hypothesis spaces such as linear models and decision trees, though several results apply more generally (Sections 5.2 and 5.3). Denote $\Omega(f)$ as a regularization term with a parameter $\lambda \in \mathbb{R}_{\geq 0}$. For example, $\Omega(\cdot)$ can be sparsity constraints or an $\ell_2$ norm. As a true risk, consider $L_{\mathcal{D}}(f) = \mathbb{E}_{\mathcal{D}}[\phi(f(x), y)]$, where $\phi : \mathbb{R}^2 \to \mathbb{R}_{\geq 0}$ is a loss function (e.g., 0–1, $\phi(f(x), y) = \mathbb{1}_{[f(x) \neq y]}$, or exponential loss, $\phi(f(x), y) = e^{-yf(x)}$).

We aim to learn a model $f^*$ from a hypothesis space $\mathcal{F}$ that minimizes the objective $obj_{\mathcal{D}}(f) = L_{\mathcal{D}}(f) + \lambda\Omega(f)$. This is approximated by minimizing the empirical objective, $\hat{obj}_S(f) = \hat{L}_S(f) + \lambda\Omega(f)$, where $\hat{L}_S(f) = \frac{1}{n}\sum \phi(f(x_i), y_i)$ is the empirical risk. Correspondingly, $\hat{f} \in \arg\min_{f \in \mathcal{F}} \hat{obj}_S(f)$ is an empirical risk minimizer (ERM). When the distribution or dataset is clear, we use the shorthand notation $L(f)$, $\hat{L}(f)$, $obj(f)$, $\hat{obj}(f)$. When $\lambda = 0$, we will directly optimize the true or empirical risks without the regularization penalty.

Following Semenova et al. (2022), given $\epsilon > 0$, we define the Rashomon set $\hat{\mathcal{R}}(\epsilon)$ as a set of near-optimal models, such that: $\hat{\mathcal{R}}(\epsilon) = \{f \in \mathcal{F} : \hat{obj}(f) \leq \hat{obj}(\hat{f}) + \epsilon\}$, where $\epsilon$ is a small tolerance parameter, typically corresponding to a slight drop in accuracy (e.g., 1–3%). Note that when regularization parameter $\lambda = 0$, then $\hat{\mathcal{R}}(\epsilon) = \{f \in \mathcal{F} : \hat{L}(f) \leq \hat{L}(\hat{f}) + \epsilon\}$. Correspondingly, we also define the true Rashomon set, based on the true objective: $\mathcal{R}(\epsilon) = \{f \in \mathcal{F} : obj(f) \leq obj(f^*) + \epsilon\}$.

With the setup in place, we analyze trustworthiness criteria at increasing levels of abstraction, moving from a single near-optimal model to the Rashomon set as a whole. When relying on a single model, a data practitioner may hope to achieve both robustness and privacy. The next section shows that achieving both simultaneously is difficult, even for interpretable or sparse models.

## 4. A Sparse Model: Private, Stable, yet Fragile

While sparsity can serve as a built-in privacy mechanism, we prove that these models are nonetheless inherently vulnerable to adversarial attacks.

## 4.1. Sparser Models are More Private and Stable

A model leaks information about its training data through the parameters or decision paths. We consider the information-theoretic perspective, where the leakage is quantified by mutual information between the learned model and the training data, denoted $I(f; S)$ (Bassily et al., 2018). Prior work shows that regularization can decrease $I(f; S)$ (Xu & Raginsky, 2017), motivating us to explore similar phenomena in sparse decision trees.

**Theorem 4.1** (Sparsity controls mutual information in a single tree). *Let $S = \{(x_i, y_i)\}_{i=1}^n$ be a dataset of $n$ i.i.d. samples from distribution $\mathcal{D}$ over $\mathcal{X} \times \mathcal{Y}$, where $\mathcal{X} = \{0, 1\}^d$ and $\mathcal{Y} = \{0, 1\}$. Let $\mathcal{F}$ be the class of binary classification decision trees with $l_f$ leaves, and let $f \in \mathcal{F}$ be a tree fit on $S$ through a possibly-random training algorithm. Then the mutual information between the learned tree $f$ and the dataset $S$ satisfies: $I(f; S) = O(l_f \log d)$.*

Theorem 4.1 shows that mutual information increases roughly linearly with the number of leaves, so sparser trees leak less and are therefore more privacy-preserving. Intuitively, fewer splits mean fewer distinct models a deterministic learner can output, which limits the entropy of $f$—an argument analogous to $\ell_0$ penalties in linear models.

Another lens on privacy is membership inference attacks, where an adversary tries to decide whether a point was in the training set. Yeom et al. (2018) show that the adversary's advantage is bounded by the model's generalization gap scaled by a constant. Since sparse models often generalize better than their non-regularized counterparts (Hastie et al., 2015; Vapnik, 1999; Shalev-Shwartz & Ben-David, 2014; Xu & Raginsky, 2017), they naturally offer stronger protection.

Note that low mutual information (high privacy) and high algorithmic stability (Bousquet & Elisseeff, 2002) are often observed together because they are a shared consequence of regularization. Indeed, Bousquet & Elisseeff (2002) shows that, for reproducing kernel Hilbert spaces, regularization strength bounds stability, meaning that stronger regularization leads to better uniform stability (see Theorem 22 (Bousquet & Elisseeff, 2002) for more details). Despite these privacy and stability benefits, a single sparse model remains inherently fragile to adversarial perturbations.

## 4.2. A Single Model is Vulnerable to Adversarial Attack

Adversarial robustness revolves around an adversarial example, which, given a sample $(x_i, y_i)$, is defined as $x_i' = x_i + \delta$ such that $f(x_i') \neq y_i$. To prevent trivial samples, perturbations are constrained within a bounded set defined by an $L_p$-norm. Formally, the set of permissible perturbations $\mathcal{S}_p = \{\delta \mid \|\delta\|_p \leq \eta\}$, where $\eta$ specifies the maximum allowed perturbation magnitude. In this work, we use $\mathcal{S}_2$ or $\mathcal{S}_\infty$ when considering continuous samples and $\mathcal{S}_0$ for binary

samples. The adversarial data $\mathcal{D}'$ can then be constructed by taking $x_i \in \mathcal{X}$ and perturbing it into the adversarial sample $x_i'$. Since in our case the true label of the perturbed sample is unknown, we use constant-in-the-ball robustness as compared to the exact-in-the-ball robustness (Gourdeau et al., 2021), which requires models to output the label of the perturbed sample rather than the label of the original sample.

In this setting, we analyze the vulnerability of a single rule list, a type of logical model composed of if-then-else statements. It is also viewed as a one-sided decision tree. Formally, a rule list with $K$ rules is defined as a quadruple $d = (d_p, \delta_p, q_0, K)$, where $d_p = (p_1, \ldots, p_K)$ is the vector of antecedents (decision split nodes on the rule list path), $\delta_p = (q_1, \ldots, q_K)$ is the vector of predictions corresponding to each decision split, and $q_0$ is the default prediction for samples that are captured by none of the antecedents (see Angelino et al. (2018) for more details). Without loss of generality, we may assume that the first prediction is 0, and it is natural to assume that every rule predicts the majority label of each point captured by that rule since this maximizes accuracy. We furthermore assume that the Boolean condition for each decision split is a conjunction of literals (e.g. $x_1 \wedge \neg x_2$ when $x_1, x_2 \in \{0, 1\}$). The following theorem characterizes the adversarial risk of such models under simple binary perturbations.

**Theorem 4.2** (Inherent vulnerability of single models). *For a dataset $S = \{(x_i, y_i)\}_{i=1}^n$ with binary features and labels, let $n_+$ be the number of data points with positive labels in $S$. Let $d = (d_p, \delta_p, q_0, K)$ be a rule list such that $q_1 = 0$, each rule predicts the majority label of the points captured by that rule, and the Boolean condition for each decision split is a conjunction of at most $l$ literals. Further, let $I$ be the smallest index $i$ such that $q_i = 1$. Let $S' = \{(x_i', y_i)\}_{i=1}^n$ be an adversarial dataset constructed by flipping up to $l$ features in each $x_i$ (i.e., an $L_0$-bounded perturbation with $\eta = l$ restricted to binary features). Let $\hat{L}$ be the 0-1 loss. If $\bar{n}_+$ is the number of positive data points captured by one of the first $I - 1$ leaves, then $\hat{L}_{S'}(d) - \hat{L}_S(d) \geq \frac{n_+ - \bar{n}_+}{n}$.*

Theorem 4.2 shows that the gap between the error and adversarial error increases with the number of positive points, meaning that a rule list with low robust error must have both low error and a high class imbalance. In other words, balanced datasets are unlikely to permit robust models. Note that this theorem provides a minimum guaranteed impact as long as the attack strength $\eta$ is above a certain threshold $l$ which is usually very small. We also note that a similar vulnerability exists for linear models under $L_2$ attack on datasets where much of the data is not well-separated, as the attack can simply push the data across the decision boundary. For both rule lists and linear models, a strong assumption on the data must be made in order for single models to be robust. While a single model is vulnerable to adversarial

perturbations, an attack designed for one model may not transfer to others in the Rashomon set. These limitations of single-model robustness naturally motivate analyzing the full set of near-optimal models next.

## 5. The Duality of the Rashomon Set

The existence of multiple, equally accurate models in the Rashomon set can present a fundamental trade-off. This section demonstrates how the existence of a large, diverse Rashomon set enables robustness and stability, but can also be exploited to create risks to data privacy.

### 5.1. Diversity Helps Robust Model Selection

In real deployments, a single deployed model is exposed to adversaries. These vulnerabilities are often discovered reactively by observing performance changes on recent data, monitoring for anomalous user or system behavior, audits that show systematic inconsistencies, red-teaming exercises, or domain-expert complaints about specific failure cases (Kreuzberger et al., 2023). One response is to apply a moving target defense, which defends against detected attacks by rotating to a different model (Amich & Eshete, 2021; He et al., 2025). In practice, many ML pipelines implement this by periodically retraining a model from scratch, which is slow and computationally expensive. Instead, we observe that the Rashomon set, whether obtained naturally during model selection or via recent estimation techniques, already provides a diverse pool of near-optimal models with comparable predictive performance. An institution can therefore respond to a detected attack by switching to a Rashomon set member that behaves differently from previously deployed models while maintaining comparable predictive performance, or proactively rotating predictions across several near-optimal models so that no single adversarial example transfers to all of them. This provides a more computationally efficient method to defend against adversarial attacks as well as a principled way to leverage diversity within the Rashomon set. We refer to the inherent robustness of this rotating model scheme as *reactive robustness*.

In this section, we show that when the Rashomon set is sufficiently diverse, reactive robustness becomes possible. Here, diversity refers to the existence of multiple near-optimal models that make meaningfully different decisions, where the precise definition of such difference is problem-dependent. For discrete hypothesis spaces such as decision trees or rule lists, we will capture diversity through prediction differences (e.g., Hamming distance between classifications). For continuous models such as linear classifiers, we will look at the diversity through geometric differences in the parameter space, such as the angle between weight vectors. Despite these specific formalizations, the takeaway on the reactive robustness is the same: if the Rashomon set

contains models that disagree with the model on the attacked points, *the adversarial vulnerability of one model will not transfer to the others*. Reactive robustness does not require identifying the attack, only that the Rashomon set contains sufficiently rich diversity so that some near-optimal models fail differently from the attacked one.

Our theoretical results below formalize this intuition. For hypothesis spaces that optimize 0–1 loss, consider a dataset $S$ with ERM $\hat{f}$. For theoretical clarity, we consider the worst case where every example in $S$ is perturbed to form an adversarial dataset, which upper-bounds the impact of any localized attack. Let $S'$ be an adversarial dataset constructed by modifying each sample $(x_i, y_i)$ to attack $\hat{f}$. Intuitively, models that are similar to $\hat{f}$ should be similarly vulnerable to this attack and thus should perform poorly on $S'$. To formalize this, we measure diversity between $f$ and $\hat{f}$ through their weighted prediction difference, $H(f, \hat{f}) = \frac{1}{n} \sum_{i=1}^{n} \mathbb{1}_{f(x_i) \neq \hat{f}(x_i)}$. This Hamming distance can be considered as a diversity measure for discrete hypothesis spaces such as decision trees, rule lists, or scoring systems. Models with higher disagreement have more distinct decision boundaries and therefore may fail differently. Then, under 0–1 loss, the triangle inequality immediately gives a robustness bound:

$$
\begin{aligned}
L_{S'}(f) = \frac{1}{n} \sum_{i=1}^{n} \mathbb{1}_{f(x_i') \neq y_i} &\leq \frac{1}{n} \sum_{i=1}^{n} \mathbb{1}_{f(x_i') \neq \hat{f}(x_i')} \\
&+ \frac{1}{n} \sum_{i=1}^{n} \mathbb{1}_{\hat{f}(x_i') \neq y_i} = H(f, \hat{f}) + L_{S'}(\hat{f}).
\end{aligned}
\tag{1}
$$

Eq (1) highlights the core mechanism behind reactive robustness: any model $f$ can only outperform the attacked model $\hat{f}$ on the adversarial dataset if it disagrees with $\hat{f}$ sufficiently often. If $H(f, \hat{f})$ is small, the bound forces $L_{S'}(f)$ to be close to $L_{S'}(\hat{f})$, so $f$ inherits the same vulnerability as $\hat{f}$. Conversely, a model that disagrees with $\hat{f}$ on the attacked points can achieve better performance on $S'$. In other words, robustness against an attack targeted at $\hat{f}$ requires diversity as models that mimic $\hat{f}$'s decisions will fail in the same way. This intuition applies broadly to any hypothesis space trained under 0-1 loss.

We can extend Eq (1) to the entire Rashomon set by considering the set-level Hamming diversity measure. If $\hat{\mathcal{R}}(\epsilon) = \{f_k\}_{k=1}^{N}$, then $\bar{d} = \frac{1}{N} \sum_{k=1}^{N} H(f_k, \hat{f})$. We can immediately upper bound the mean adversarial loss of the Rashomon set in terms of the set-level diversity:

$$
\frac{1}{N} \sum_{k=1}^{N} L_{S'}(f_k) \leq \bar{d} + L_{S'}(\hat{f}).
\tag{2}
$$

Next, we provide more detailed theoretical evidence in the linear setting, where diversity, captured through geometry,

similarly enables robustness through non-transferability of adversarial attacks.

Consider the hypothesis space of linear models $f(x) = w^T x$ where $w \in \mathbb{R}^p$ and let $\hat{w}_S$ be the ERM model for some dataset $S$. As before, let $S'$ be an adversarial dataset generated from $S$ using the $L_2$ norm and targeted to maximize the classification error of $\hat{w}_S$. That is, $x' = x + \delta$, where $\|\delta\|_2 \leq \eta$ and $\delta$ is chosen to make $y \cdot (\hat{w}_S)^T x'$ as small or negative as possible (e.g., $\delta \approx -\eta y \frac{\hat{w}_S}{\|\hat{w}_S\|_2}$). Here, we use the angle between weight vectors as our measure of diversity. We will show that models whose weight vectors form a larger angle have more distinct decision boundaries and are therefore less likely to share the same adversarial vulnerabilities. In this sense, angular distance plays the same role for linear models that prediction disagreement played for 0-1 loss in the discussion above. For margin-based losses (i.e. losses that depend on functional margin $yf(x)$, such as exponential loss), we can compute the loss of an arbitrary linear model on the adversarial dataset as follows:

**Theorem 5.1** (Risk on adversarial dataset). *Suppose that* $\hat{L}_S(w) = \frac{1}{n} \sum_{i=1}^{n} \phi(y_i \cdot w^T x_i)$ *where $\phi$ is a loss that is a function of the margin $(y_i f(x_i))$. For an $L_2$ attack on the optimal model $\hat{w}_S$ with budget $\eta$, the loss of $w$ on the adversarial dataset $S'$ is $\hat{L}_{S'}(w) = \frac{1}{n} \sum_{i=1}^{n} \phi(y_i \cdot w^T x_i - \eta \|w\|_2 \cos(w, \hat{w}_S))$.*

Proofs of all theorems and corollaries are provided in the appendix. For any reasonable objective, the loss $\phi$ should decrease with the margin. Thus, we intuitively have that, as the angle between a given model $w$ and the optimal model increases (as measured in $\cos(w, \hat{w}_S)$), the margin should decrease faster, so the loss should increase faster. The next corollary formalizes this intuition for the exponential loss.

**Corollary 5.2.** *Suppose that we have the exponential loss $\phi(y \cdot w^T x) = e^{-y \cdot w^T x}$. Then, for any unit weights $w_1$ and $w_2$ satisfying $\cos(w_1, \hat{w}_S) > \cos(w_2, \hat{w}_S)$, we have that $\frac{\hat{L}_{S'}(w_1)}{\hat{L}_S(w_1)} > \frac{\hat{L}_{S'}(w_2)}{\hat{L}_S(w_2)}$. In other words, the adversarial attack is most effective on models more similar to $\hat{w}_S$.*

Further, as the strength of the adversarial attack grows, models that are of greater angle with the optimal model will eventually surpass the performance of lower angle models.

**Corollary 5.3.** *Suppose that $\phi(\cdot)$ is decreasing with respect to the margin, and let $w_1$ and $w_2$ be unit weights so that $\cos(w_1, \hat{w}_S) > \cos(w_2, \hat{w}_S)$. Then, for large enough $\eta$ (e.g. when $\eta \to \infty$), $\hat{L}_{S'}(w_1) > \hat{L}_{S'}(w_2)$.*

Corollaries 5.2 and 5.3 show that models that make a greater angle with the optimal model are more robust to adversarial attacks. Therefore, if the Rashomon set is diverse enough to contain these models, they will perform well on both datasets $S$ and $S'$. We do not claim that any single diversity metric guarantees robustness against all adaptive attacks;

rather, our results show that without disagreement aligned to the attack geometry, adversarial transferability is unavoidable. Importantly, this does not assume identification of the attack mechanism. For example, if the Rashomon set is diverse only through parallel shifts of the boundary, keeping $w$ aligned with $\hat{w}$, then this diversity will likely not guarantee the existence of a robust model in the Rashomon set. Note that we observe similar results in the setting of least-squares regression as we discuss in Appendix C.

### 5.2. Rashomon Set is Stable to Small Distribution Shifts

After establishing reactive robustness benefits, we now show that the Rashomon set as a whole remains stable under a small distribution shift. More specifically, we consider the scenario where the underlying data distribution changes slightly under covariate shift. If this shift is small, models that were good under the original distribution remain good under the new distribution, within a slightly relaxed performance threshold as we show in the next theorem.

**Theorem 5.4** (Rashomon set is robust under small distribution shift). *Consider a bounded loss function $\phi$, $\phi \in [0, 1]$, and two data distributions $\mathcal{D}$ and $\mathcal{D}'$, such that $\mathcal{D}(x) \neq \mathcal{D}'(x)$ and $\mathcal{D}(y|x) = \mathcal{D}'(y|x)$. If $KL(\mathcal{D}\|\mathcal{D}') \leq \frac{\epsilon^2}{8}$, then if a function $f$ is in the true Rashomon set for the data distribution $\mathcal{D}$, $f \in \mathcal{R}_{z \sim \mathcal{D}}(\frac{\epsilon}{2})$, it is also in the true Rashomon set for the data distribution $\mathcal{D}'$, $f \in \mathcal{R}_{z \sim \mathcal{D}'}(\epsilon)$.*

Theorem 5.4 shows that the Rashomon set is stable: if a model performs well on one data distribution, it will still perform well, relative to the new best model, even after a small shift in the distribution. This does not guarantee absolute accuracy, as shifts that increase task difficulty may degrade performance across all models. But the theorem ensures the drop is smooth because the set of good models changes gradually, so models that were strong before the shift remain among the best options after the shift. In other words, one doesn't need to start the search for a good model all over again and can simply look for it within the existing Rashomon set.

Interestingly, the same stability occurs empirically, if the two datasets, $S$ and $S'$, differ by only a small number of data points. In this case, the empirical Rashomon sets defined on these two datasets are similar. Specifically, any model belonging to the Rashomon set for one dataset is guaranteed to belong to the Rashomon set of the other dataset if we slightly increase the tolerance threshold by an amount proportional to the number of differing samples ($K/n$).

**Theorem 5.5** (Two Rashomon sets constructed on neighboring datasets are indistinguishable). *For 0-1 loss, let $S$ and $S'$ be two datasets, each of size $n$. Let $\hat{\mathcal{R}}_S(\epsilon) := \{f \in \mathcal{F} | \hat{obj}(f, S) \leq \hat{obj}(\hat{f}, S) + \epsilon\}$ and $\hat{\mathcal{R}}_{S'}(\epsilon) := \{f \in \mathcal{F} | \hat{obj}(f, S') \leq \hat{obj}(\hat{f}', S') + \epsilon\}$. Suppose $S$ and $S'$ differ*

in at most $K$ samples. Then $\hat{\mathcal{R}}_S(\epsilon) \subseteq \hat{\mathcal{R}}_{S'}(\epsilon + \frac{2K}{n})$ and $\hat{\mathcal{R}}_{S'}(\epsilon) \subseteq \hat{\mathcal{R}}_S(\epsilon + \frac{2K}{n})$.

Theorem 5.5 establishes the stability of the Rashomon set when the dataset undergoes minor modifications, such as adding, removing, or changing a few examples. Note that while Theorem 4.2 suggests that such perturbations may decrease the absolute performance of the optimal model(s), Theorems 5.4 and 5.5 show that the Rashomon set as a whole can remain stable. We also verify this dataset-level stability empirically. Using four datasets and pre-computed Rashomon sets, we modify $K$ samples in each dataset and recompute the Rashomon set on the modified data. When we increase the Rashomon tolerance from $\varepsilon$ to $\varepsilon + 2K/n$, the new Rashomon sets remain highly overlapping with the original. Even under a 6% dataset modification, more than 80% of the models remain, and for smaller perturbations (e.g., 2%), the sets are nearly identical (Figure 11). This confirms that small perturbations to the dataset do not radically change the pool of near-optimal models, providing practical evidence for Rashomon-set stability.

For an auditor, our observation means that if empirical Rashomon sets remain overlapping under small dataset perturbations, the institution's choice among near-optimal interpretable models is reproducible as re-running the learning pipeline on new data will not radically change the pool of plausible models. The near-invariance in Theorem 5.5 also has implications for privacy as we discuss next.

### 5.3. Larger Rashomon Sets Leak More Information

While the diversity can be a powerful defense, it also creates new risks. Imagine an organization that shares several of its top-performing interpretable models to promote transparency or comply with regulations. The release of each model on its own may seem safe, especially given privacy-preserving properties of sparse models. But together, they can reveal more than intended. In this section, we explore how an adversary might combine information from the released set of "safe" models to construct a privacy attack that is capable of reconstructing sensitive information from the original training data. We consider the Rashomon set for an arbitrary model class and focus on binary classification.

**Theorem 5.6** (Distribution bound for random ensembles from the Rashomon set). *Let $\mathcal{R}(\epsilon)$ be the Rashomon set trained on $S$ containing $N$ models $\{f_1, \ldots, f_N\}$, and define $p(x) := P(y = 1|x)$ based on $S$. Let $\mu(x)$ be the mean and $\sigma^2(x)$ be the variance of predictions in the Rashomon set for input $x$. Suppose we sample $m$ models without replacement from $\mathcal{R}(\epsilon)$, where $m \leq N$, with sample indices $\Pi = (\pi_1, \pi_2, ..., \pi_m)$ chosen uniformly without replacement from $\{1, 2, ...N\}$. Define the ensemble prediction as $q_\Pi(x) := \frac{1}{m} \sum_{k=1}^{m} f_{\pi_k}(x)$. Then, the expected TV distance between $p(x)$ and the ensemble pre-*diction $q_\Pi(x)$ is bounded by: $E_\Pi[\mathrm{TV}(p(x)||q_\Pi(x))] \leq \sqrt{(p(x) - \mu(x))^2 + \frac{(N-m)\sigma^2(x)}{(N-1)m}}$.

*If we furthermore know that $f_k(x) \in [\delta, 1 - \delta]$ for each model $k$, then we have the following bound on the KL divergence: $E_\Pi[KL(p(x)||q_\Pi(x))] \leq \frac{(p(x)-\mu(x))^2 + \frac{(N-m)\sigma^2(x)}{(N-1)m}}{\delta(1-\delta)}$.*

Theorem 5.6 reveals a privacy trade-off induced by Rashomon multiplicity. Even if each model in the Rashomon set is sparse and therefore individually privacy-preserving, releasing many such models increases aggregate leakage. The ensemble of models provides a more accurate approximation of the data distribution, as reflected by the decreasing KL divergence between $p(x)$ and $q_\Pi(x)$ (Figure 5). In this sense, larger Rashomon sets offer more "views" of the data, which collectively reveal more information about the underlying distribution.

It is important to distinguish this distributional leakage from *membership privacy*, which concerns whether an individual training example can be inferred from the released models. The distributional leakage does not automatically imply stronger membership-inference risk for individual data points. Membership privacy concerns whether releasing a model reveals whether a specific example was contained in the training set. Theorem 5.5 shows that Rashomon sets constructed on neighboring datasets (differing in only a few samples) are nearly identical as any model in one set is very likely to appear in the other. For large datasets, releasing the Rashomon set therefore reveals almost the same information for two neighboring datasets, limiting an adversary's ability to infer the presence or absence of individual records. This aligns with the intuition behind differential privacy.

Finally, we note that diversity in the Rashomon set also has implications for robustness, both for individual models and for ensembles. In Appendix C, Theorem C.5 upper bounds the variance of predictions $\sigma^2(x)$ in terms of the set-level diversity $\bar{d}$. When combined with Equation 2, this gives a tradeoff between privacy and robustness that is explicitly controlled by the diversity of the Rashomon set. Furthermore, Theorem C.6 shows that forming an ensemble from sufficiently diverse models can improve robustness by reducing the chance that an adversarial perturbation affects all models simultaneously. This form of robustness is different from the reactive robustness studied in Section 5.1. Instead of switching to a new near-optimal model after an attack is detected, ensemble methods combine predictions from multiple diverse models to limit the transferability of the attack. This provides a mechanism for robustness that complements the reactive robustness view in Section 5.1. Importantly, the trade-off we discussed still holds because releasing many diverse models, whether individually or as part of an ensemble, increases distributional information leakage as characterized by Theorem 5.6. Next, we focus

on our empirical findings.

# 6. Experiments

We now present experimental results on sparse decision trees using TreeFARMS (Xin et al., 2022) to examine how diversity within the Rashomon set affects adversarial robustness, information leakage, and the resulting trade-off.

## 6.1. Diversity Benefits Adversarial Robustness

Given that sparse decision trees are inherently interpretable, we consider white-box attacks. We enumerate all possible perturbations that lead to incorrect predictions for the optimal tree in the Rashomon set, and study how other trees in the Rashomon set respond to these adversarial examples (Kantchelian et al., 2016). We report the adversarial score defined as the accuracy on the adversarial dataset and the distance to the optimal tree calculated by the Hamming distance between the classification patterns. Given a dataset $S$, a classification pattern $C_f(S)$ is an $n$-tuple of predicted labels $C_f(S) = (f(x_1), f(x_2), \ldots, f(x_n))$. We use these classification patterns to measure diversity in (1) in Section 5.1 as well as in our experiments.

Figure 2 shows a strong positive trend between a tree's distance from the optimal model and its adversarial score. Trees with classification patterns similar to the optimal tree (bottom left) are more vulnerable to adversarial examples, while those that differ more in their predictions (top right) are more robust. This suggests that diversity of the Rashomon set benefits adversarial robustness, as a subset of trees can maintain high adversarial accuracy even when others fail.

As more and more models are released from the Rashomon set, adversaries may design their attacks to target multiple released models simultaneously, which could in theory circumvent the benefits of diversity. However, this is difficult in practice for two reasons. Firstly, it is currently computationally expensive to attack multiple diverse models at the same time. Secondly, there might not always exist attacks that are simultaneously valid against many diversely chosen models. In Appendix E.3, we experimentally demonstrate the feasibility of reactive robustness even under multi-model attacks for the hypothesis space of decision trees.

## 6.2. Diversity Accelerates Information Leakage

While our theoretical analysis quantifies privacy leakage via mutual information, computing it exactly is often infeasible in practice. In our experiments, we therefore adopt a more tractable and widely used proxy: reconstruction error from a dataset reconstruction attack. This approach reflects an adversary's ability to recover training data from released models and serves as an operational measure of privacy risk.

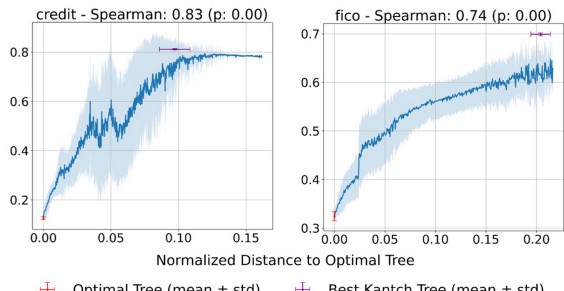

*Figure 2.* Adversarial score (accuracy) of trees in the Rashomon set vs. their distance to the optimal tree. Results are aggregated over five folds. The optimal trees (in red) are attacked. The most robust trees (in purple) are far from the optimal tree. Trees with the same distance to optimal trees are grouped, and mean and standard deviation of their adversarial score are shown as line plots with shaded uncertainty. See Figure 6 for more plots.

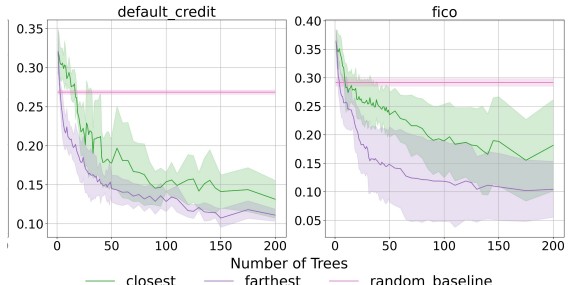

*Figure 3.* Comparison of reconstruction error between different selection strategies. The random baseline randomly guesses the feature values for each data point. See Figure 9 for more plots.

DRAFT (Ferry et al., 2024) can reconstruct the dataset used by a random forest by solving a constraint programming problem. We use the reconstruction error from this attack as a proxy for information leakage, where a lower reconstruction error indicates greater leakage. Note that this attack is stronger than membership inference attacks since reconstructing individual training points allows an adversary to answer membership queries as a special case.

Following the setup in Ferry et al. (2024), we sample 100 data points to train a Rashomon set. Trees are then sequentially selected from the Rashomon set and passed to DRAFT. We run DRAFT multiple times as more trees are added. Specifically, DRAFT is run after each additional tree from 1 to 50, every 5 trees from 50 to 150, and again at 175 and 200 trees. In total, we consider up to 200 trees from the Rashomon set. We run this process five times with different random seeds. We consider two strategies to select trees. By default, the first tree selected is the optimal model. The *closest* strategy then iteratively selects the tree whose classification pattern has the smallest Hamming distance to that of the optimal tree. The *farthest* strategy greedily selects the tree whose pattern has the largest Hamming distance from those of the previously selected trees.

Figure 3 shows that the "farthest" strategy (in purple) has lower reconstruction error, indicating greater information leakage. The "closest" strategy (in green) has better privacy, as it has a higher reconstruction error. These results suggest that more diversity in the Rashomon set and disclosing more diverse models lead to increased privacy risk.

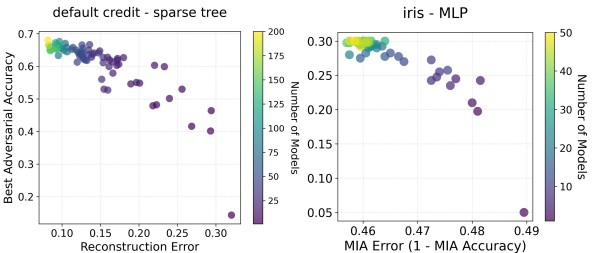

*Figure 4.* Attack error vs. adversarial accuracy for ensembles constructed with different numbers of evenly sampled trees (left) or multi-layer perceptrons (right) from the Rashomon set.

### 6.3. Robustness-Privacy Tradeoffs under Rashomon Set

To study the relationship between robustness and privacy, we designed a new strategy to sample trees from the Rashomon set. We first sort trees by Hamming distance of their classification patterns to the optimal tree, then select trees at evenly spaced intervals. For instance, given a Rashomon set of 1000 trees, an ensemble of 100 trees would include every 10th tree in the sorted list. To evaluate robustness-privacy tradeoffs, we report both the reconstruction error and the best adversarial accuracy among the selected trees.

Figure 4 (left) shows the reconstruction error versus adversarial accuracy for ensembles constructed from selected trees. Each point represents an ensemble of a specific size, with color indicating the number of trees included. When only a limited number of models are released, the privacy is preserved but the models are not diverse enough to avoid an adversarial attack targeted on the optimal trees. As more trees are selected, robustness improves but at the cost of greater information leakage. This result aligns with our previous findings and provides direct evidence of a robustness-privacy trade-off when releasing the Rashomon set in the wild. Figure 4 (right) shows the trade-off for a multi-layer perceptron. Additional experimental details, figures, and discussion are provided in Appendix D and E.5.

### 6.4. Operationalizing the Trade-off

Given the robustness–privacy trade-off curves in Figures 4 and 23, we consider how they can be used to guide disclosure decisions. Globally, the average slope of these curves quantifies the exchange rate of the trade-off: the marginal robustness gain per unit of privacy loss. In this context, the steepness of the slope serves as a direct measure of efficiency. Steeper negative slopes (e.g., COMPAS, $\approx -2$)

indicate a high-efficiency regime, where the Rashomon set provides substantial reactive robustness for a minimal privacy cost; while flatter slopes (e.g., FICO, $\approx -1$) indicate diminishing returns, where the privacy loss outweighs the robustness gains.

While there is no universal approach to exploring the Pareto frontier, in Appendix E.6 we discuss a "privacy-first" selection rule that serves as a starting point for balancing privacy and robustness using a hybrid strategy. As shown in Figure 23, this approach adapts to the data landscape, prioritizing efficiency for robust tasks (e.g., Bank Marketing) where gains saturate early, while enforcing strict safety limits on vulnerable datasets (e.g., COMPAS) where reconstruction error collapses faster. This confirms that while slope efficiency determines viability, hard safety constraints can ultimately dictate the deployment scale. Ultimately, this framework demonstrates how stakeholders may operationalize the trade-off, tailoring model deployment to balance their specific efficiency goals with organizational risk tolerance.

## 7. Conclusions and Implications

This work shows that diversity within the Rashomon set enables reactive robustness through a rotating model system while increasing privacy leakage when many near-optimal models are exposed, because the same differences between models that block adversarial attack transferability also yield complementary views of the underlying data. As a result, trustworthiness is shaped not only by the deployed model but by the broader set of alternatives generated during development. These findings motivate a shift from single-model governance to set-level governance. Organizations may benefit from retaining a diverse Rashomon set internally to support robustness and reproducibility while limiting external disclosure of near-optimal models to reduce privacy risk. Rather than releasing entire Rashomon sets, institutions may instead identify the disclosure "sweet spot" or disclose representative models, balancing transparency with privacy. Taken together, these observations motivate future work on diversity-aware disclosure policies and practical tools for auditing Rashomon sets in real deployments.

## Impact Statement

We study adversarial robustness and data privacy through the lens of the Rashomon set. Our analysis shows that model diversity can support robustness improvements while also amplifying privacy risks under specific attack scenarios, indicating that the disclosure of multiple near-optimal models may itself carry trustworthiness implications in high-stakes settings.

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

## A. Summary of Results

For ease of navigation, we provide a summary of the results presented within this work in Table 2.

| Hypothesis Space | Discrete Models (Trees, Rule Lists) | Linear Models | Ensembles |
|---|---|---|---|
| **Diversity Measure** | Hamming Distance | Cosine Similarity | Pairwise Correlation |
| **Single Model Vulnerability** | Thm 4.2 | Section 4.2 Discussion | N/A |
| **Reactive Robustness** | Eq 2 | Thm 5.1, Cor 5.2, 5.3 | Thm C.6, C.7 |
| **Leakage** | Thm 5.6 | Thm 5.6 | Thm 5.6 |
| **Stability** | Thm 5.4, 5.5 | Thm 5.4 | Thm 5.4, 5.5 |

*Table 2.* Summary of the results contained in this work for each hypothesis space.

## B. Proofs for Theoretical Results in Section 4

In this appendix, we provide proofs for the theoretical results presented in Section 4.

**Theorem 4.2** (Inherent vulnerability of single models). *For a dataset $S = \{(x_i, y_i)\}_{i=1}^{n}$ with binary features and labels, let $n_+$ be the number of data points with positive labels in $S$. Let $d = (d_p, \delta_p, q_0, K)$ be a rule list such that $q_1 = 0$, each rule predicts the majority label of the points captured by that rule, and the Boolean condition for each decision split is a conjunction of at most $l$ literals. Further, let $I$ be the smallest index $i$ such that $q_i = 1$. Let $S' = \{(x_i', y_i)\}_{i=1}^{n}$ be an adversarial dataset constructed by flipping up to $l$ features in each $x_i$ (i.e., an $L_0$-bounded perturbation with $\eta = l$ restricted to binary features). Let $\hat{L}$ be the 0-1 loss. If $\bar{n}_+$ is the number of positive data points captured by one of the first $I - 1$ leaves, then $\hat{L}_{S'}(d) - \hat{L}_S(d) \geq \frac{n_+ - \bar{n}_+}{n}$.*

*Proof.* Let $a_k$ and $b_k$ denote the number of negative (zero) and positive points captured by rule $k$. Then, note that all data points captured by rules with index at least $I$ are vulnerable to the adversarial attack. Thus, we have the following:

$$K_+ = \sum_{k=1}^{K} b_k$$

$$\hat{L}_S(d) = \frac{1}{n} \left[ \sum_{k=1}^{I-1} b_k + \sum_{k=I}^{K} (a_k q_k + b_k(1 - q_k)) \right]$$

$$\hat{L}_{S'}(d) \geq \frac{1}{n} \left[ \sum_{k=1}^{I-1} b_k + \sum_{k=I}^{K} (a_k + b_k) \right]$$

$$\bar{K}_+ = \sum_{k=1}^{I-1} b_k.$$

From here, note that $q_k = \mathbb{1}_{a_k \leq b_k}$ since each rule predicts the majority label. Then, we can write

$$(K_+ - \bar{K}_+) + (n\hat{L}_S(d) - \bar{K}_+) = \sum_{k=I}^{K} b_k + \sum_{k=I}^{K} (a_k q_k + b_k(1 - q_k))$$

$$= \sum_{k=I}^{K} (a_k + b_k) + \sum_{k=I}^{K} (1 - q_k)(b_k - a_k)$$

$$\leq \sum_{k=I}^{K} (a_k + b_k)$$

$$\leq n\hat{L}_{S'}(d) - \bar{K}_+,$$

where $(1 - q_k)(b_k - a_k) = \mathbb{1}_{a_k > b_k}(b_k - a_k) \leq 0$. Therefore, we get that $\hat{L}_{S'}(d) - \hat{L}_S(d) \geq \frac{K_+ - \bar{K}_+}{n}$. $\qquad\square$

The theorem above states that there is an inherent cost to robustness when attacking a single model. Therefore, having the whole Rashomon set can be useful for model selection if adversarial attacks are expected.

**Theorem 4.1** (Sparsity controls mutual information in a single tree). *Let $S = \{(x_i, y_i)\}_{i=1}^n$ be a dataset of $n$ i.i.d. samples from distribution $\mathcal{D}$ over $\mathcal{X} \times \mathcal{Y}$, where $\mathcal{X} = \{0,1\}^d$ and $\mathcal{Y} = \{0,1\}$. Let $\mathcal{F}$ be the class of binary classification decision trees with $l_f$ leaves, and let $f \in \mathcal{F}$ be a tree fit on $S$ through a possibly-random training algorithm. Then the mutual information between the learned tree $f$ and the dataset $S$ satisfies: $I(f; S) = O(l_f \log d)$.*

*Proof.* The mutual information between the model $f$ and the dataset $S$ can be expressed as $I(f; S) = H(f) - H(f|S)$, and the upper bound $I(f; S) \leq H(f)$. It is well known that entropy is bounded by the logarithm of the size of the alphabet, so we next bound the size of the hypothesis space, $|\mathcal{F}|$.

Let $T(l_f)$ denote the number of binary trees with $l_f$ leaves. A full binary tree with $l_f$ leaves has exactly $l_f - 1$ internal nodes and $2l_f - 1$ total nodes. It is well known that the number of such tree shapes is the $(l_f - 1)$-st Catalan number, so $T(l_f) = C_{l_f - 1}$.

To bound $|\mathcal{F}|$, note that each decision tree can be obtained by taking a binary tree, assigning splits to each internal node, and assigning a prediction to each leaf. Each internal node has at most $d$ features to split on, and it selects a direction (less than or greater than). Each leaf node can likewise choose to either predict $0$ or $1$. Thus, since there are $l_f - 1$ internal nodes with at most $2d$ choices at each internal node, and there are $l_f$ leaves with $2$ choices at each leaf, we have that $|\mathcal{F}| \leq T(l_f) \cdot (2d)^{l_f - 1} 2^{l_f} = C_{l_f - 1} \cdot (2d)^{l_f - 1} 2^{l_f}$. Since $C_{l_f - 1}$ grows at most exponentially in $l_f$, we have that $\log |\mathcal{F}| = O(l_f \log d)$. Thus, $I(f; S) = O(l_f \log d)$.

$\square$

# C. Proofs for Theoretical Results in Section 5

In this appendix, we provide proofs for the theoretical results presented in Section 5.

## C.1. Proof for Theorem 5.4

We state and prove Theorem 5.4 below. Note that for the purposes of the next theorem, the model belonging to the Rashomon set is based on the risk $L_{\mathcal{D}}$, meaning that $\mathcal{R}(\epsilon) = \{f \in \mathcal{F} : L(f) \leq L(f^*) + \epsilon\}$, where $f^*$ is optimal model.

**Theorem 5.4** (Rashomon set is robust under small distribution shift). *Consider a bounded loss function $\phi$, $\phi \in [0,1]$, and two data distributions $\mathcal{D}$ and $\mathcal{D}'$, such that $\mathcal{D}(x) \neq \mathcal{D}'(x)$ and $\mathcal{D}(y|x) = \mathcal{D}'(y|x)$. If $KL(\mathcal{D}\|\mathcal{D}') \leq \frac{\epsilon^2}{8}$, then if a function $f$ is in the true Rashomon set for the data distribution $\mathcal{D}$, $f \in \mathcal{R}_{z \sim \mathcal{D}}(\frac{\epsilon}{2})$, it is also in the true Rashomon set for the data distribution $\mathcal{D}'$, $f \in \mathcal{R}_{z \sim \mathcal{D}'}(\epsilon)$.*

*Proof.* Given the bounded loss function $\phi(f(x), y)$ for a data point $z = (x, y)$, we will overload definition of the loss and also consider $\phi(f, z) : \mathcal{F} \times (\mathcal{X} \times \mathcal{Y}) \to [0, 1]$. Let $f_{\mathcal{D}}^* = \arg\inf_{g \in \mathcal{F}} \mathbb{E}_{z \sim \mathcal{D}}[\phi(g, z)]$ and $f_{\mathcal{D}'}^* = \arg\inf_{g \in \mathcal{F}} \mathbb{E}_{z \sim \mathcal{D}'}[\phi(g, z)]$ be optimal models for distributions $\mathcal{D}$ and $\mathcal{D}'$ respectively. Also, let $d_{TV}(\mathcal{D}, \mathcal{D}')$ be the total variation distance defined as:

$$d_{TV}(\mathcal{D}, \mathcal{D}') = \sup_{B \in \mathcal{B}} |\Pr_{\mathcal{D}}[B] - \Pr_{\mathcal{D}'}[B]| = \frac{1}{2} \int |p_{\mathcal{D}}(z) - p_{\mathcal{D}'}(z)| dz,$$

where $\mathcal{B}$ is the set of measurable subsets under $\mathcal{D}$ and $\mathcal{D}'$. Then since $\phi(\cdot, \cdot) \in [0, 1]$, we have $|\phi(g, z) - 1/2| \leq 1/2$ for any

$g \in \mathcal{F}$. Thus:

$$
\begin{aligned}
|\mathbb{E}_{z \sim \mathcal{D}} \phi(g, z) - \mathbb{E}_{z \sim \mathcal{D}'} \phi(g, z)| &= \left| \int_z \phi(g, z)(p_{\mathcal{D}}(z) - p_{\mathcal{D}'}(z)) dz \right| \\
&= \left| \int_z (\phi(g, z) - \frac{1}{2})(p_{\mathcal{D}}(z) - p_{\mathcal{D}'}(z)) dz \right| \\
&\leq \int_z |\phi(g, z) - \frac{1}{2}| \cdot |p_{\mathcal{D}}(z) - p_{\mathcal{D}'}(z)| dz \\
&\leq \int_z \frac{1}{2} \cdot |p_{\mathcal{D}}(z) - p_{\mathcal{D}'}(z)| dz \\
&= \frac{1}{2} \int |p_{\mathcal{D}}(z) - p_{\mathcal{D}'}(z)| dz \\
&= d_{TV}(\mathcal{D}, \mathcal{D}')
\end{aligned}
\tag{3}
$$

Note that (3) holds for $g = f$ and $g = f_{\mathcal{D}}^*$ as well.

By Pinsker's inequality $d_{TV}(\mathcal{D}, \mathcal{D}') \leq \sqrt{\frac{1}{2} KL(\mathcal{D} \| \mathcal{D}')}$. Since $f \in \mathcal{R}_{z \sim \mathcal{D}}(\frac{\epsilon}{2})$ and $KL(\mathcal{D} \| \mathcal{D}') \leq \frac{\epsilon^2}{8}$, we have that $d_{TV}(\mathcal{D}, \mathcal{D}') \leq \sqrt{\frac{1}{2} KL(\mathcal{D} \| \mathcal{D}')} \leq \sqrt{\frac{1}{2} \frac{\epsilon^2}{8}} = \sqrt{\frac{\epsilon^2}{16}} = \frac{\epsilon}{4}$. Therefore, we can bound the expected risks' difference for distribution $\mathcal{D}'$:

$$
\begin{aligned}
\mathbb{E}_{z \sim \mathcal{D}'} \phi(f, z) - \mathbb{E}_{z \sim \mathcal{D}'} \phi(f_{\mathcal{D}'}^*, z) &= (\mathbb{E}_{z \sim \mathcal{D}} \phi(f, z) - \mathbb{E}_{z \sim \mathcal{D}} \phi(f_{\mathcal{D}}^*, z)) \\
&\quad + (\mathbb{E}_{z \sim \mathcal{D}'} \phi(f, z) - \mathbb{E}_{z \sim \mathcal{D}} \phi(f, z)) \\
&\quad + (\mathbb{E}_{z \sim \mathcal{D}} \phi(f_{\mathcal{D}}^*, z) - \mathbb{E}_{z \sim \mathcal{D}'} \phi(f_{\mathcal{D}'}^*, z)) \\
&\leq \frac{\epsilon}{2} \quad \left(\text{Since } f \in \mathcal{R}_{z \sim \mathcal{D}}\left(\frac{\epsilon}{2}\right)\right) \\
&\quad + |\mathbb{E}_{z \sim \mathcal{D}'} \phi(f, z) - \mathbb{E}_{z \sim \mathcal{D}} \phi(f, z)| + (\mathbb{E}_{z \sim \mathcal{D}} \phi(f_{\mathcal{D}}^*, z) - \mathbb{E}_{z \sim \mathcal{D}'} \phi(f_{\mathcal{D}'}^*, z)) \\
&\leq \frac{\epsilon}{2} + |\mathbb{E}_{z \sim \mathcal{D}'} \phi(f, z) - \mathbb{E}_{z \sim \mathcal{D}} \phi(f, z)| \\
&\quad + (\mathbb{E}_{z \sim \mathcal{D}} \phi(f_{\mathcal{D}}^*, z) - \mathbb{E}_{z \sim \mathcal{D}'} \phi(f_{\mathcal{D}}^*, z)) \quad (\text{Since } f_{\mathcal{D}'}^* \text{ minimizes for } \mathcal{D}') \\
&\leq \frac{\epsilon}{2} + |\mathbb{E}_{z \sim \mathcal{D}'} \phi(f, z) - \mathbb{E}_{z \sim \mathcal{D}} \phi(f, z)| + |\mathbb{E}_{z \sim \mathcal{D}} \phi(f_{\mathcal{D}}^*, z) - \mathbb{E}_{z \sim \mathcal{D}'} \phi(f_{\mathcal{D}}^*, z)| \\
&\leq \frac{\epsilon}{2} + d_{TV}(\mathcal{D}, \mathcal{D}') + d_{TV}(\mathcal{D}, \mathcal{D}') \quad (\text{Using result from (3)}) \\
&= \frac{\epsilon}{2} + 2d_{TV}(\mathcal{D}, \mathcal{D}') \leq \frac{\epsilon}{2} + 2\left(\frac{\epsilon}{4}\right) \quad (\text{Since } d_{TV}(\mathcal{D}, \mathcal{D}') \leq \frac{\epsilon}{4}) \\
&= \frac{\epsilon}{2} + \frac{\epsilon}{2} = \epsilon.
\end{aligned}
$$

Therefore, $f \in \mathcal{R}_{z \sim \mathcal{D}'}(\epsilon)$. $\qquad \square$

Note that for ridge regression, we can derive a more specific condition on the total variation distance as we show in Lemma C.4.

### C.2. Proof for Theorem 5.5

We state and prove Theorem 5.5 below. Note that while the theorem is proved for the objective-defined Rashomon set, the statement holds for the risk defined Rashomon set when the regularization parameter is zero.

**Theorem 5.5** (Two Rashomon sets constructed on neighboring datasets are indistinguishable). *For 0-1 loss, let $S$ and $S'$ be two datasets, each of size $n$. Let $\hat{\mathcal{R}}_S(\epsilon) := \{f \in \mathcal{F} | \hat{obj}(f, S) \leq \hat{obj}(\hat{f}, S) + \epsilon\}$ and $\hat{\mathcal{R}}_{S'}(\epsilon) := \{f \in \mathcal{F} | \hat{obj}(f, S') \leq \hat{obj}(\hat{f}', S') + \epsilon\}$. Suppose $S$ and $S'$ differ in at most $K$ samples. Then $\hat{\mathcal{R}}_S(\epsilon) \subseteq \hat{\mathcal{R}}_{S'}(\epsilon + \frac{2K}{n})$ and $\hat{\mathcal{R}}_{S'}(\epsilon) \subseteq \hat{\mathcal{R}}_S(\epsilon + \frac{2K}{n})$.*

*Proof.* For any model $f \in \mathcal{F}$, the difference in 0-1 loss due to change of the dataset from $S$ to $S'$ that differ in at most $K$ samples is at most $\frac{K}{n}$. Hence,

$$|o\hat{b}j_S(f) - o\hat{b}j_{S'}(f)| = |\hat{L}_S(f) - \hat{L}_{S'}(f)| \leq \frac{K}{n}. \tag{4}$$

By definition, since $\hat{f}$ is ERM, $o\hat{b}j_S(\hat{f}) \leq o\hat{b}j_S(\hat{f}')$. Plugging in $\hat{f}'$ in (4), we get that: $o\hat{b}j_S(\hat{f}') \leq o\hat{b}j_{S'}(\hat{f}') + \frac{K}{n}$. Therefore, we get that

$$o\hat{b}j_{S'}(\hat{f}') \geq o\hat{b}j_S(\hat{f}') - \frac{K}{n} \geq o\hat{b}j_S(\hat{f}) - \frac{K}{n}. \tag{5}$$

Next, we show for the empirical Rashomon sets that $\hat{\mathcal{R}}_S(\epsilon) \subseteq \hat{\mathcal{R}}_{S'}(\epsilon + \frac{2K}{n})$. For any model $f \in \hat{\mathcal{R}}_S(\epsilon)$, by definition, $o\hat{b}j_S(f) \leq o\hat{b}j_S(\hat{f}) + \epsilon$. Based on (4) we get that,

$$o\hat{b}j_{S'}(f) \leq o\hat{b}j_S(f) + \frac{K}{n} \leq o\hat{b}j_S(\hat{f}) + \epsilon + \frac{K}{n}. \tag{6}$$

Finally, combining (5) and (6) together, we see that

$$o\hat{b}j_{S'}(f) - o\hat{b}j_{S'}(\hat{f}') \leq o\hat{b}j_S(\hat{f}) + \epsilon + \frac{K}{n} - o\hat{b}j_S(\hat{f}) + \frac{K}{n} = \epsilon + \frac{2K}{n}, \tag{7}$$

which means that $f \in \hat{\mathcal{R}}_{S'}(\epsilon + \frac{2K}{n})$ and correspondingly, $\hat{\mathcal{R}}_S(\epsilon) \subseteq \hat{\mathcal{R}}_{S'}(\epsilon + \frac{2K}{n})$. Following similar logic we can show that $\hat{\mathcal{R}}_{S'}(\epsilon) \subseteq \hat{\mathcal{R}}_S(\epsilon + \frac{2K}{n})$ yielding the statement of the theorem. $\qquad\square$

### C.3. Proofs for Theorem 5.1 and Corollaries 5.2 and 5.3

We state and prove Theorem 5.1 as well as two corollaries from it below.

**Theorem 5.1.** *Suppose that $\hat{L}_S(w) = \frac{1}{n}\sum_{i=1}^{n} \phi(y_i \cdot w^T x_i)$ where $\phi$ is a loss that is a function of the margin $(y_i f(x_i))$. For an $L_2$ attack on the optimal model $\hat{w}_S$ with budget $\eta$, the loss of $w$ on the adversarial dataset $S'$ is $\hat{L}_{S'}(w) = \frac{1}{n}\sum_{i=1}^{n} \phi(y_i \cdot w^T x_i - \eta \|w\|_2 \cos(w, \hat{w}_S))$.*

*Proof.* For each sample $x_i$, under the adversarial attack, we have that $x_i' = x_i + \delta_i$ where $\delta_i = -\eta y_i \frac{\hat{w}_S}{\|\hat{w}_S\|_2}$. Then, the margin for a given model $w$ on the adversarial input $x_i$ is:

$$
\begin{aligned}
y_i \cdot w^T x_i' &= y_i \cdot w^T (x_i + \delta_i) \\
&= y_i \cdot w^T x_i + y_i \cdot w^T \delta_i \\
&= y_i \cdot w^T x_i + y_i \cdot w^T \left(-\eta y_i \frac{\hat{w}_S}{\|\hat{w}_S\|_2}\right) \\
&= y_i \cdot w^T x_i - \eta y_i^2 \frac{w^T \hat{w}_S}{\|\hat{w}_S\|_2} \\
&= y_i \cdot w^T x_i - \eta \frac{w^T \hat{w}_S}{\|\hat{w}_S\|_2} \\
&= y_i \cdot w^T x_i - \eta \|w\|_2 \cos(w, \hat{w}_S).
\end{aligned}
$$

Therefore, we get that $\hat{L}_{S'}(w) = \frac{1}{n}\sum_{i=1}^{n} \phi(y_i \cdot w^T x_i - \eta \|w\|_2 \cos(w, \hat{w}_S))$. $\qquad\square$

In the next corollary, we show that if the model $w^T x$ is closer to the ERM model, then the adversarial attack has more effect on it in terms of the loss.

**Corollary C.1.** *Suppose that we have the exponential loss $\phi(y \cdot w^T x) = e^{-y \cdot w^T x}$. Then, for any unit weights $w_1$ and $w_2$ satisfying $\cos(w_1, \hat{w}_S) > \cos(w_2, \hat{w}_S)$, we have that $\frac{\hat{L}_{S'}(w_1)}{\hat{L}_S(w_1)} > \frac{\hat{L}_{S'}(w_2)}{\hat{L}_S(w_2)}$. In other words, the adversarial attack is most effective on models more similar to $\hat{w}_S$.*

*Proof.* For any unit vector $w$, we have by Theorem 5.1 that

$$\hat{L}_{S'}(w) = \frac{1}{n}\sum_{i=1}^{n}\exp(-y_i \cdot w^T x_i + \eta\cos(w,\hat{w}_S)) = \exp(\eta\cos(w,\hat{w}_S))\hat{L}_S(w).$$

Then, we can write

$$\begin{aligned}
\frac{\hat{L}_{S'}(w_1)}{\hat{L}_S(w_1)} &= \frac{\exp(\eta\cos(w_1,\hat{w}_S))\hat{L}_S(w_1)}{\hat{L}_S(w_1)} \\
&= \exp(\eta\cos(w_1,\hat{w}_S)) \\
&> \exp(\eta\cos(w_2,\hat{w}_S)) \\
&= \frac{\hat{L}_{S'}(w_2)}{\hat{L}_S(w_2)}
\end{aligned}$$

$\square$

Our next corollary makes this point even more explicit in the case of a strong attack, showing that if the diversity within the Rashomon set is higher in terms of angular distance, then models with higher adversarial risk can exist.

**Corollary C.2.** *Suppose that $\phi(\cdot)$ is decreasing with respect to the margin, and let $w_1$ and $w_2$ be unit weights so that $\cos(w_1,\hat{w}_S) > \cos(w_2,\hat{w}_S)$. Then, for large enough $\eta$ (e.g. when $\eta \to \infty$), $\hat{L}_{S'}(w_1) > \hat{L}_{S'}(w_2)$.*

*Proof.* Since $S$ is finite, it suffices to show that, for each $i$, there exists some $N_i$ such that

$$\phi(y_i \cdot w_1^T x_i - \eta\cos(w_1,\hat{w}_S)) > \phi(y_i \cdot w_2^T x_i - \eta\cos(w_2,\hat{w}_S))$$

for all $\eta \ge N_i$. However, this is true since $\eta\cos(w_1,\hat{w}_S)$ is arbitrarily larger than $\eta\cos(w_2,\hat{w}_S)$ as $\eta \to \infty$, so $y_i \cdot w_1^T x_i - \eta\cos(w_1,\hat{w}_S) < y_i \cdot w_2^T x_i - \eta\cos(w_2,\hat{w}_S)$ for sufficiently large $\eta$. $\square$

Next, we focus on a different loss function for linear models, specifically the least-squares loss used in regression tasks.

## C.4. Proof for Theorem C.3

For our second setting, we consider least-squares regression, where now the loss is $\phi(f(x),y) = (f(x)-y)^2$. Our hypothesis space is still linear models $w^T x$. Then the Rashomon set is an ellipsoid $(w-\hat{w})^T\mathbb{E}[xx^T](w-\hat{w}) \le \epsilon$, where the singular values of the matrix $\mathbb{E}[xx^T]$ determine its shape. Let $TV(\mathcal{D},\mathcal{D}')$ be the total variation distance between two distributions, then under label shift, we can bound the minimum singular value $\sigma_{min}$ of the matrix $\mathbb{E}[xx^T]$ as follows:

**Theorem C.3.** *Let $\mathcal{D}$ and $\mathcal{D}'$ be two data distributions in $\mathcal{X} \times \mathcal{Y}$ such that $\mathcal{D}(x) = \mathcal{D}'(x)$ but $\mathcal{D}(y|x) \ne \mathcal{D}'(y|x)$. Furthermore, suppose that $y \in [a,b]$ for all $y \in \mathcal{Y}$ and suppose that $\|x\|_2 \le C$ for all $x \in \mathcal{X}$. Then, there exists some model in both true Rashomon sets $\mathcal{R}_\mathcal{D}(\epsilon)$ and $\mathcal{R}_{\mathcal{D}'}(\epsilon)$ if*

$$TV(\mathcal{D},\mathcal{D}') \le \frac{2\sqrt{\epsilon}}{(b-a)\cdot C}\cdot\sqrt{\sigma_{min}(E[xx^T])}.$$

To prove Theorem C.3, we use Lemma C.4 which we prove below.

**Lemma C.4.** *Suppose that $y \in [a,b]$ for all $y \in \mathcal{Y}$, and suppose that $\|x\|_2 \le C$ for all $x \in \mathcal{X}$. Furthermore, suppose that we undergo a distribution shift to $\mathcal{D}'$ where $\mathcal{D}(x) = \mathcal{D}'(x)$ but $\mathcal{D}(y|x) \ne \mathcal{D}'(y|x)$. Then, if $\hat{w}_{\mathcal{D}'}$ is the optimal linear model for $\mathcal{D}'$ under the least-squares objective, we have that*

$$\left\|E_{x\sim\mathcal{X}}[xx^T](\hat{w}_\mathcal{D} - \hat{w}_{\mathcal{D}'})\right\|_2 \le (b-a)\cdot C \cdot TV(\mathcal{D},\mathcal{D}').$$

*Proof.* For a data distribution $\mathcal{D}$, the optimal weights for a linear model under the least-squares objective is

$$\hat{w}_\mathcal{D} = (E_{x\sim\mathcal{X}}[xx^T])^{-1}E_{(x,y)\sim\mathcal{D}}[xy]. \tag{8}$$

Then, we can write

$$\left\|E[xx^T](\hat{w}_\mathcal{D} - \hat{w}_{\mathcal{D}'})\right\|_2 = \left\|E_{(x,y)\sim\mathcal{D}}[xy] - E_{(x,y)\sim\mathcal{D}'}[xy]\right\|_2$$

$$= \left\|\int_x x\mathcal{D}(x)\int_y y(\mathcal{D}(y|x) - \mathcal{D}'(y|x))\right\|_2$$

$$= \left\|\int_x x\mathcal{D}(x)\int_y \left(y - \frac{a+b}{2}\right)(\mathcal{D}(y|x) - \mathcal{D}'(y|x))\right\|_2$$

$$\leq \int_x \|x\|_2 \mathcal{D}(x)\int_y \left|y - \frac{a+b}{2}\right||\mathcal{D}(y|x) - \mathcal{D}'(y|x)|$$

$$\leq \frac{b-a}{2}\cdot\int_x \|x\|_2 \mathcal{D}(x)\int_y |\mathcal{D}(y|x) - \mathcal{D}'(y|x)|$$

$$\leq \frac{(b-a)\cdot C}{2}\cdot\int_x \mathcal{D}(x)\int_y |\mathcal{D}(y|x) - \mathcal{D}'(y|x)|$$

$$= \frac{(b-a)\cdot C}{2}\cdot\int_x\int_y |\mathcal{D}(x,y) - \mathcal{D}'(x,y)|$$

$$= (b-a)\cdot C\cdot TV(\mathcal{D}, \mathcal{D}')$$

as desired. $\qquad\square$

Now we prove Theorem C.3, which bounds the total variation distance with the value of the minimum singular value of the expected data matrix.

*Proof.* From Theorem 10 of (Semenova et al., 2022), we know that $\mathcal{R}_\mathcal{D}(\epsilon)$ is the ellipsoid described by the equation

$$(w - \hat{w}_\mathcal{D})^T \frac{E[xx^T]}{\epsilon}(w - \hat{w}_\mathcal{D}) \leq 1.$$

We have a similar equation for $\mathcal{R}_{\mathcal{D}'}(\epsilon)$. Equivalently, if $M = (\frac{E[xx^T]}{\epsilon})^{\frac{1}{2}}$ and $S(0,1)$ is the unit ball centered at the origin, then $\mathcal{R}_\mathcal{D}(\epsilon) = M^{-1}S(0,1) + \hat{w}_\mathcal{D} = \{w\colon \|M(w - \hat{w}_\mathcal{D})\|_2 \leq 1\}$. Consider $\bar{w} = \frac{\hat{w}_\mathcal{D} + \hat{w}_{\mathcal{D}'}}{2}$, then

$$\|M(\bar{w} - \hat{w}_\mathcal{D})\|_2 = \left\|\frac{M(\hat{w}_{\mathcal{D}'} - \hat{w}_\mathcal{D})}{2}\right\|_2 = \frac{1}{2}\|M(\hat{w}_{\mathcal{D}'} - \hat{w}_\mathcal{D})\|_2.$$

Next we will show that $\bar{w} \in \mathcal{R}_\mathcal{D}(\epsilon)$. If the bound on the total variation distance in the theorem assumption holds and given Lemma C.4, we have that:

$$\|M(\hat{w}_{\mathcal{D}'} - \hat{w}_\mathcal{D})\|_2 = \left\|\frac{1}{\sqrt{\epsilon}}E[xx^T]^{-\frac{1}{2}}E[xx^T](\hat{w}_{\mathcal{D}'} - \hat{w}_\mathcal{D})\right\|_2$$

$$\leq \frac{1}{\sqrt{\epsilon}}\left\|E[xx^T]^{-\frac{1}{2}}\right\|_2 \left\|E[xx^T](\hat{w}_{\mathcal{D}'} - \hat{w}_\mathcal{D})\right\|_2$$

$$= \frac{1}{\sqrt{\epsilon}\sqrt{\sigma_{min}(E[xx^T])}}\left\|E[xx^T](\hat{w}_{\mathcal{D}'} - \hat{w}_\mathcal{D})\right\|_2$$

$$\leq \frac{(b-a)\cdot C}{\sqrt{\epsilon}\sqrt{\sigma_{min}(E[xx^T])}}\cdot TV(\mathcal{D}, \mathcal{D}')$$

$$\leq 2.$$

Therefore,

$$\|M(\bar{w} - \hat{w}_\mathcal{D})\|_2 = \frac{1}{2}\|M(\hat{w}_{\mathcal{D}'} - \hat{w}_\mathcal{D})\|_2 \leq 1,$$

which means that $\bar{w} \in \mathcal{R}_\mathcal{D}(\epsilon)$. A similar argument shows that $\bar{w} \in \mathcal{R}_{\mathcal{D}'}(\epsilon)$, which proves the theorem. $\qquad\square$

Theorem C.3 means that for a model to remain robust across larger distribution shifts (i.e., increased $TV(\mathcal{D}, \mathcal{D}'))$, the data's second moment matrix $E[\mathbf{x}\mathbf{x}^\top]$ must have a higher minimum singular value, $\sigma_{\min}(E[\mathbf{x}\mathbf{x}^\top])$. A relatively high $\sigma_{\min}(E[\mathbf{x}\mathbf{x}^\top])$, particularly when the overall spectrum of singular values for $E[\mathbf{x}\mathbf{x}^\top]$ is well-conditioned contributes to the "roundness" of the Rashomon set. In turn, this roundness can be viewed as a form of structural diversity, meaning that the set contains models whose parameters reflect more uniform importance across different feature directions. Therefore, such diverse sets are more likely to contain models that can be selected for their robustness.

## C.5. The variance of predictions of the Rashomon set is upper-bounded by set-level diversity

**Theorem C.5** (Variance upper bound)**.** *Let* $\mathcal{R}(\epsilon)$ *be the Rashomon set trained on* $S = \{(x_i, y_i)\}_{i=1}^n$ *containing* $N$ *models* $\{f_1, \ldots, f_N\}$, *and suppose that each model outputs a binary prediction in* $\{0, 1\}$. *Let* $\sigma^2(x)$ *be the variance of predictions in the Rashomon set for some input* $x$, *and let* $\bar{d}$ *be the set-level Hamming diversity measure for* $\mathcal{R}(\epsilon)$. *Then,*

$$\frac{1}{n}\sum_{i=1}^n \sigma^2(x_i) \leq \bar{d}.$$

*Proof.* We first show that $\frac{1}{n}\sum_{i=1}^n \sigma^2(x_i) = \frac{1}{2N^2}\sum_{i,j} H(f_i, f_j)$. For any fixed $x$, since $f_k(x) \in \{0, 1\}$, we have that $\mathbb{1}_{f_i(x) \neq f_j(x)} = (f_i(x) - f_j(x))^2$, so:

$$\frac{1}{N^2}\sum_{i,j} \mathbb{1}_{f_i(x) \neq f_j(x)} = \frac{1}{N^2}\sum_{i,j}(f_i(x) - f_j(x))^2$$

$$= \frac{1}{N^2}\sum_{i,j}\left(f_i(x)^2 - 2f_i(x)f_j(x) + f_j(x)^2\right)$$

$$= \frac{2}{N}\sum_i f_i(x)^2 - \frac{2}{N^2}\sum_{i,j} f_i(x)f_j(x)$$

$$= 2\mu(x) - 2\mu(x)^2 = 2\mu(x)(1 - \mu(x)) = 2\sigma^2(x)$$

where $\mu(x) = \frac{1}{N}\sum_k f_k(x)$, and we used the fact that $f_k(x)^2 = f_k(x)$ for binary predictions. By averaging over the dataset, we see that

$$\frac{1}{n}\sum_{i=1}^n \sigma^2(x_i) = \frac{1}{2N^2}\sum_{i,j} H(f_i, f_j).$$

Lastly, we have by the triangle inequality:

$$\frac{1}{N^2}\sum_{i,j} H(f_i, f_j) \leq \frac{1}{N^2}\sum_{i,j}(H(f_i, \hat{f}) + H(f_j, \hat{f})) = \frac{2}{N}\sum_k H(f_k, \hat{f}) = 2\bar{d}.$$

$\square$

We note that this is the same variance used in Theorem 5.6, and that the bound there weakens as $\sigma^2(x)$ increases. This means that, as the diversity of the Rashomon set increases, there is a greater risk of data leakage.

## C.6. Diverse ensemble of models from the Rashomon set is more adversarially robust

The proofs in Section 5.1 show that models that are diverse (e.g. rely on different logic) are less vulnerable to the adversarial attack of the optimal model. We can generalize this intuition to an ensemble. More specifically, we consider a majority-vote ensemble of models from the Rashomon set. First, we consider independent models in Theorem C.6. If the models are sufficiently diverse, their failures on a given adversarial input might be treated as largely independent events. We relax this assumption to allow weak correlations between models in Theorem C.7.

**Theorem C.6** (Independent ensemble)**.** *Let* $\{f_1, f_2, ..., f_k\}$ *be a subset of models in the Rashomon set where* $k$ *is odd (to prevent ties). Let* $\delta : \mathbb{R}^d \to \mathbb{R}^d$ *be a function that takes in a data point and outputs a (possibly random) perturbation for that point. For a random data point* $(x, y)$ *drawn from the distribution* $\mathcal{D}$, *let* $Z_i = \mathbf{1}_{[f_i(x+\delta(x)) \neq y]}$ *be a random variable indicating whether model* $f_i$ *predicts the perturbed data point incorrectly. Let* $p_i = Pr_{(x,y)}(Z_i = 1)$. *Assume that there*

*exists $p < \frac{1}{2}$ such that $p_i \leq p$ for all $i$. Let $S_k = \sum_{i=1}^{k} Z_i$ be the random count of individual models that the attack fools on a single input. Since $k$ is odd, the probability $Pr_{(x,y)}(S_k \geq k/2)$ is exactly the chance that at least half of the $k$ models are wrong, which means the majority-vote ensemble is also wrong on that adversarially-perturbed input. Suppose $Z_1, Z_2, \ldots, Z_k$ are independent. Then,*

$$Pr_{(x,y)}(S_k \geq k/2) \leq e^{-kD_{KL}(\frac{1}{2}||p)}.$$

*Proof.* Based on the Chernoff bound, we can get, for $t > 0$,

$$Pr(S_k \geq k/2) = Pr(e^{tS_k} \geq e^{tk/2}) \leq \mathbb{E}[e^{tS_k}]e^{-tk/2}.$$

Since $Z_i's$ are independent,

$$\mathbb{E}[e^{tS_k}] = \mathbb{E}[e^{t(\sum_{i=1}^{k} Z_i)}] = \mathbb{E}[\Pi_{i=1}^{k} e^{tZ_i}] = \Pi_{i=1}^{k}\mathbb{E}[e^{tZ_i}].$$

Since $Z_i$ is a Bernoulli variable, $e^{tZ_i} = 1 + (e^t - 1)Z_i$. Then,

$$\mathbb{E}[e^{tS_k}] = \Pi_{i=1}^{k}\mathbb{E}[1 + (e^t - 1)Z_i] = \Pi_{i=1}^{k}(1 + (e^t - 1)\mathbb{E}[Z_i]) = \Pi_{i=1}^{k}(1 + (e^t - 1)p_i).$$

Since $p_i \leq p, \forall i$,

$$\mathbb{E}[e^{tS_k}] = \Pi_{i=1}^{k}(1 + (e^t - 1)p_i) \leq \Pi_{i=1}^{k}(1 + (e^t - 1)p) = (1 + (e^t - 1)p)^k.$$

Then, $\mathbb{E}[e^{tS_k}]e^{-tk/2} \leq (1 - p + pe^t)^k e^{-tk/2}$. Let $h(t) = \ln(1 - p + pe^t) - \frac{t}{2}$. Then,

$$(1 - p + pe^t)^k e^{-tk/2} = e^{kh(t)}.$$

Since the bound holds for any $t > 0$, let's find the value of $t$ that gives us the tightest bound.

$$\frac{dh}{dt} = \frac{pe^t}{1 - p + pe^t} - \frac{1}{2}.$$

Set it to 0, we get $t^* = \ln\frac{1-p}{p}, e^{t^*} = \frac{1-p}{p}$. Then we know,

$$h(t^*) = \ln(1 - p + 1 - p) - \frac{1}{2}\ln\frac{1-p}{p}$$

$$= \ln 2 + \ln(1 - p) - \frac{1}{2}\ln(1 - p) + \frac{1}{2}\ln p$$

$$= \ln 2 + \frac{1}{2}\ln((1 - p)p)$$

$$= \ln(2\sqrt{(1 - p)p}).$$

Now, we know $\mathbb{E}[e^{tS_k}]e^{-tk/2} \leq (1 - p + pe^{t^*})^k e^{-t^* k/2} = e^{kh(t^*)}$. Since $p \in [0, 1/2]$, $(1 - p)p \in [0, 1/4]$ and $h(t^*) \in (-\infty, 0)$. Hence, $e^{kh(t^*)}$ decreases as $k$ increases.

Also, note that $KL(\frac{1}{2}||p) = -\ln(2\sqrt{(1 - p)p})$. Therefore, $e^{kh(t^*)} = e^{-k \cdot KL(\frac{1}{2}||p)}$. $\square$

Theorem C.6 shows that even if individual models have a non-trivial probability of being fooled by an attack, the probability that a majority-vote ensemble of these models fails decreases exponentially with the size of the ensemble.

We can also consider the dependent case, when there is correlation between models in the ensemble to get similar bounds:

**Theorem C.7** (Dependent ensemble). *Let $\{f_1, f_2, ..., f_k\}$ be a subset of models in the Rashomon set, $k > 2$, and $k$ is odd (to prevent ties). Let $\delta : \mathbb{R}^d \to \mathbb{R}^d$ be a function that takes in a data point and outputs a (possibly random) perturbation for that point. For a random data point $(x, y)$ drawn from the distribution $\mathcal{D}$, let $Z_i = \mathbf{1}_{[f_i(x+\delta(x))\neq y]}$ be a random variable indicating whether model $f_i$ predicts the perturbed data point incorrectly. Let $p_i = Pr(Z_i = 1)$. Assume that there exists $p < \frac{1}{2}$ such that $p_i \leq p$ for all $i$. Let $S_k = \sum_{i=1}^{k} Z_i$ be the random count of individual models that the attack fools on a single input. Since $k$ is odd, the probability $Pr(S_k \geq k/2)$ is exactly the chance that at least half of the $k$ models are wrong, which means the majority-vote ensemble is also wrong on that adversarially-perturbed input. Assume that pairwise correlation between models is bounded, $|corr(Z_i, Z_j)| \leq \rho$ for all $i \neq j$, where $0 \leq \rho < 1$. Then*

$$Pr\left(S_k \geq k/2\right) \leq \frac{p(1-p)\left[1 + (k-1)\rho\right]}{p(1-p)\left[1 + (k-1)\rho\right] + k\left(\frac{1}{2} - p\right)^2}.$$

*Proof.* The variance of $Z_i$ is at most $p(1 - p)$. The variance of $S_k$ is at most $kp(1 - p) + k(k - 1)\rho \cdot p(1 - p) = p(1 - p)[k + k(k - 1)\rho]$. The expectation of $S_k$ is at most $kp$. Then by Cantelli's Inequality,

$$
\begin{aligned}
P(S_k \geq k/2) = P(S_k - E[S_k] &\geq k/2 - E[S_k]) \\
&\leq P(S_k - E[S_k] \geq k(0.5 - p)) \\
&\leq \frac{\sigma^2}{\sigma^2 + k^2(\frac{1}{2} - p)^2} \\
&\leq \frac{p(1 - p)[k + k(k - 1)\rho]}{p(1 - p)[k + k(k - 1)\rho] + k^2(\frac{1}{2} - p)^2} \\
&= \frac{p(1 - p)[1 + (k - 1)\rho]}{p(1 - p)[1 + (k - 1)\rho] + k(\frac{1}{2} - p)^2}
\end{aligned}
$$

$\square$

As $k \to \infty$, we can see that the bound approaches $\frac{1}{1 + \frac{(\frac{1}{2} - p)^2}{p(1 - p) \cdot \rho}}$, which increases with both $\rho$ and $p$. Thus, as long as the subset of models sampled from the Rashomon set is diverse and therefore decorrelated, an ensemble created from these models is robust to perturbations.

## C.7. Proof for Theorem 5.6

**Theorem 5.6** (Distribution bound for random ensembles from the Rashomon set). *Let $\mathcal{R}(\epsilon)$ be the Rashomon set trained on $S$ containing $N$ models $\{f_1, \ldots, f_N\}$, and define $p(x) := P(y = 1|x)$ based on $S$. Let $\mu(x)$ be the mean and $\sigma^2(x)$ be the variance of predictions in the Rashomon set for input $x$. Suppose we sample $m$ models without replacement from $\mathcal{R}(\epsilon)$, where $m \leq N$, with sample indices $\Pi = (\pi_1, \pi_2, ..., \pi_m)$ chosen uniformly without replacement from $\{1, 2, ...N\}$. Define the ensemble prediction as $q_\Pi(x) := \frac{1}{m} \sum_{k=1}^m f_{\pi_k}(x)$. Then, the expected TV distance between $p(x)$ and the ensemble prediction $q_\Pi(x)$ is bounded by: $E_\Pi[\mathrm{TV}(p(x)||q_\Pi(x))] \leq \sqrt{(p(x) - \mu(x))^2 + \frac{(N-m)\sigma^2(x)}{(N-1)m}}$.*

*If we furthermore know that $f_k(x) \in [\delta, 1 - \delta]$ for each model $k$, then we have the following bound on the KL divergence: $E_\Pi[KL(p(x)||q_\Pi(x))] \leq \frac{(p(x) - \mu(x))^2 + \frac{(N-m)\sigma^2(x)}{(N-1)m}}{\delta(1 - \delta)}.$*

*Proof.* We focus on the proof for the KL divergence since the proof for the TV distance follows nearly identically. The KL divergence between two Bernoulli distributions with parameters $p = p(x)$ and $q = q_\Pi(x)$ is given by $KL(p||q) = p \log \frac{p}{q} + (1 - p) \log \frac{1-p}{1-q}$. Using the Chi-squared divergence upper bound on KL divergence ($KL(p||q) \leq \chi^2(p||q)$), we get:

$$
KL(p||q) \leq \frac{(p - q)^2}{q(1 - q)}, \quad \text{provided } q \in (0, 1).
$$

By assumption, $f_k(x) \in [\delta, 1 - \delta]$ for all $k = 1, \ldots, N$ and some $\delta \in (0, 1/2)$. This implies the ensemble prediction $q_\Pi(x) = \frac{1}{m} \sum_{k=1}^m f_{\pi_k}(x)$ also lies in $[\delta, 1 - \delta]$. Therefore, the denominator $q_\Pi(x)(1 - q_\Pi(x))$ is bounded below: $q_\Pi(x)(1 - q_\Pi(x)) \geq \delta(1 - \delta) > 0$.

Applying this bound to the KL inequality we get that:

$$
KL(p(x)||q_\Pi(x)) \leq \frac{(p(x) - q_\Pi(x))^2}{q_\Pi(x)(1 - q_\Pi(x))} \leq \frac{(p(x) - q_\Pi(x))^2}{\delta(1 - \delta)}.
$$

Now, we take the expectation over the random sampling $\Pi$ of $m$ models:

$$
\mathbb{E}_\Pi[KL(p(x)||q_\Pi(x))] \leq \mathbb{E}_\Pi\left[\frac{(p(x) - q_\Pi(x))^2}{\delta(1 - \delta)}\right] = \frac{1}{\delta(1 - \delta)} \mathbb{E}_\Pi[(p(x) - q_\Pi(x))^2].
$$

Let $\mu(x) = E_\Pi[q_\Pi(x)]$ be the mean prediction over the entire Rashomon set. We decompose the expected squared error term:

$$\mathbb{E}_\Pi[(p(x) - q_\Pi(x))^2] = \mathbb{E}_\Pi[(p(x) - \mu(x) + \mu(x) - q_\Pi(x))^2]$$
$$= \mathbb{E}_\Pi[(p(x) - \mu(x))^2 + (\mu(x) - q_\Pi(x))^2 + 2(p(x) - \mu(x))(\mu(x) - q_\Pi(x))]$$
$$= (p(x) - \mu(x))^2 + \mathbb{E}_\Pi[(\mu(x) - q_\Pi(x))^2]$$
$$= (p(x) - \mu(x))^2 + Var_\Pi(q_\Pi(x)),$$

where because of the linearity of expectation we used that $\mathbb{E}_\Pi[\mu(x) - q_\Pi(x)] = \mu(x) - \mathbb{E}_\Pi[q_\Pi(x)] = \mu(x) - \mu(x) = 0$ and $\mathbb{E}_\Pi[(\mu(x) - q_\Pi(x))^2]$ is the variance of the sample mean $q_\Pi(x)$, denoted as $Var_\Pi(q_\Pi(x))$. For sampling $m$ items without replacement from a finite population of size $N$ with variance $\sigma^2(x) = \frac{1}{N} \sum_{k=1}^{N} (f_k(x) - \mu(x))^2$, the variance of the sample mean is:

$$Var_\Pi(q_\Pi(x)) = \frac{\sigma^2(x)}{m} \left( \frac{N - m}{N - 1} \right) = \frac{(N - m)\sigma^2(x)}{(N - 1)m}.$$

Substituting this variance back into the expression for the expected squared error:

$$\mathbb{E}_\Pi[(p(x) - q_\Pi(x))^2] = (p(x) - \mu(x))^2 + \frac{(N - m)\sigma^2(x)}{(N - 1)m},$$

which gives us the inequality for the expected KL divergence:

$$\mathbb{E}_\Pi[KL(p(x)||q_\Pi(x))] \leq \frac{(p(x) - \mu(x))^2 + \frac{(N-m)\sigma^2(x)}{(N-1)m}}{\delta(1 - \delta)}.$$

For the TV distance, we note that

$$\mathbb{E}_\Pi[\mathrm{TV}(p(x), q_\Pi(x))] = E_\Pi[|p(x) - q_\Pi(x)|] \leq \sqrt{E_\Pi[(p(x) - q_\Pi(x))^2]}$$

by Jensen's Inequality. We can then follow the argument above. $\qquad\square$

From the theorem above we know that as $m$ increases towards $N$, the variance term in the numerator decreases (since $N - m \geq 0$), thus reducing the upper bound on the expected distance between the empirical probability $p(x)$ and the ensemble prediction $q_\Pi(x)$. Note that this bound is not limited to the Rashomon set of decision trees. It applies to the Rashomon set of other hypothesis spaces. Figure 5 illustrates the empirical validity of Theorem 5.6 across the five synthetic distributions described above. As the size of the ensemble increases (x-axis), the average KL divergence between the true distribution $p(x)$ and the ensemble prediction $q_\Pi(x)$ decreases monotonically (y-axis). This confirms that disclosing larger subsets of the Rashomon set allows an adversary to approximate the underlying data distribution with increasing precision, therefore accelerating information leakage. While the convergence rate varies depending on the complexity of the ground-truth distribution, the downward trend is consistent.

## D. Experimental Setup and Results

### D.1. Computation Resources

We performed experiments on a 2.7Ghz (768GB RAM 48 cores) Intel Xeon Gold 6226 processor. Each model is trained individually on one core per dataset. We requested 32GB memory for each parallel run.

### D.2. Dataset

We present results for 6 datasets: four are from the UCI Machine Learning Repository (Dua & Graff, 2017) (Adult, Bank, Credit, and Diabetes), a recidivism dataset (COMPAS) (Larson et al., 2016), and the Fair Isaac (FICO) credit risk dataset (FICO et al., 2018) used for the Explainable ML Challenge. We predict which individuals are arrested within two years of release on the COMPAS dataset, and whether an individual will default on a loan for the FICO dataset. The detailed experimental setups are provided in Appendix D.3. The summary of datasets is in Table 3.

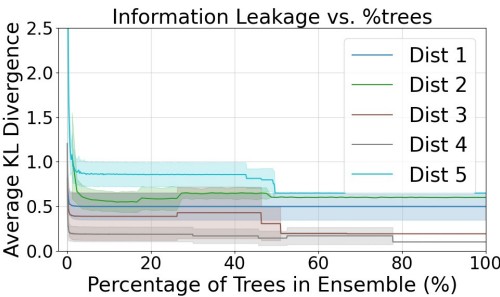

*Figure 5.* KL divergence between $p(x)$ and $q_\Pi(x)$ decreases as more trees are released into the ensemble.

*Table 3.* Summary of datasets used in experiments.

| Dataset | #Rows | #Features |
|---|---|---|
| Adult | 48,843 | 20 |
| Bank Marketing | 45,212 | 48 |
| COMPAS | 7,207 | 15 |
| Default Credit | 29,987 | 22 |
| Diabetes | 45,716 | 82 |
| FICO | 9,872 | 30 |

## D.3. More Experimental Results

In this appendix, we present additional experimental results for the robustness and privacy analysis.

*Table 4.* Summary of parameters for adversarial robustness experiment and number of trees averaged over five folds.

| Dataset | Adult | Bank | COMPAS | Credit | Diabetes | FICO |
|---|---|---|---|---|---|---|
| Rashomon adder $\epsilon$ | 0.01 | 0.02 | 0.01 | 0.01 | 0.04 | 0.01 |
| Average # Trees | 595508.0 | 1525531.2 | 846640.0 | 127601.4 | 99961.6 | 300473.2 |

**Diversity in the Rashomon Set Benefits Adversarial Robustness**
**Collection and Setup:** We ran this experiment on 6 datasets. Since TreeFARMS takes binary input, we binarized all datasets using the threshold guessing technique proposed in (McTavish et al., 2022) with n_estimators = 30, max_depth = 2. To run TreeFARMS, we set regularization to 0.01 and depth_budget to 5. The value of $\epsilon$ is tuned to ensure that the constructed Rashomon set contains a sufficient number of trees, usually more than 100,000. The exact $\epsilon$ values and the corresponding number of trees for each dataset are provided in Table 4. We adopt the algorithm proposed in (Kantchelian et al., 2016) to attack the optimal tree, the tree with the lowest objective value. The attack uses the $\mathcal{S}_\infty$ set with $\eta = 0.1$ as defined in section 5.1. We report the performance of other trees in the Rashomon set on the adversarial dataset. For visualization purposes, we group the trees based on their prediction patterns on the validation set and measure their Hamming distance to the prediction pattern of the optimal tree.

**Results:** Figure 6, a more comprehensive version of Figure 2, shows that trees with classification patterns similar to the optimal tree (bottom left corner) are more vulnerable to adversarial examples, while those that differ more in their predictions (top right corner) are more robust. The reported Spearman correlation coefficients are all positive, and for 4 out of 6 datasets, the coefficients are above 0.7, indicating a strong positive correlation between diversity and adversarial robustness. The relatively lower Spearman correlation for the Adult and Bank datasets may be due to the limited diversity within their Rashomon sets, which might be caused by the data distributions.

Figure 7 shows a scatterplot of each tree's distance to the optimal tree versus its adversarial score. As we can see, for the COMPAS, Credit, Diabetes, and FICO datasets, we observe a strong positive trend. While for the Adult and Bank datasets, some trees with zero distance from the attacked tree still perform well on the adversarial examples, an overall increasing trend remains visible in the scatter plots, supporting our main conclusion.

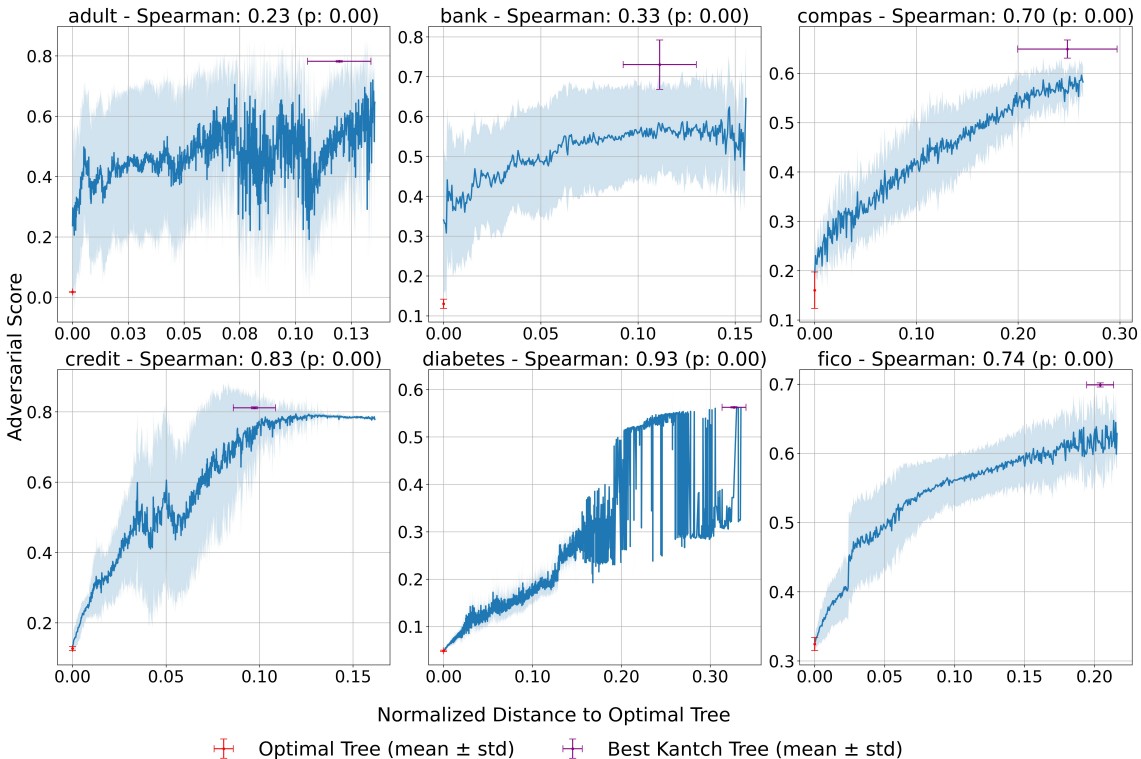

*Figure 6.* Adversarial score of trees in Rashomon set vs. their distance to optimal tree. Results are aggregated over five folds. The optimal trees (in red) are attacked. The most robust trees (in purple) are far from optimal tree. Trees with the same distance to optimal trees are grouped, and mean and standard deviation of their adversarial score are shown as line plots with shaded uncertainty.

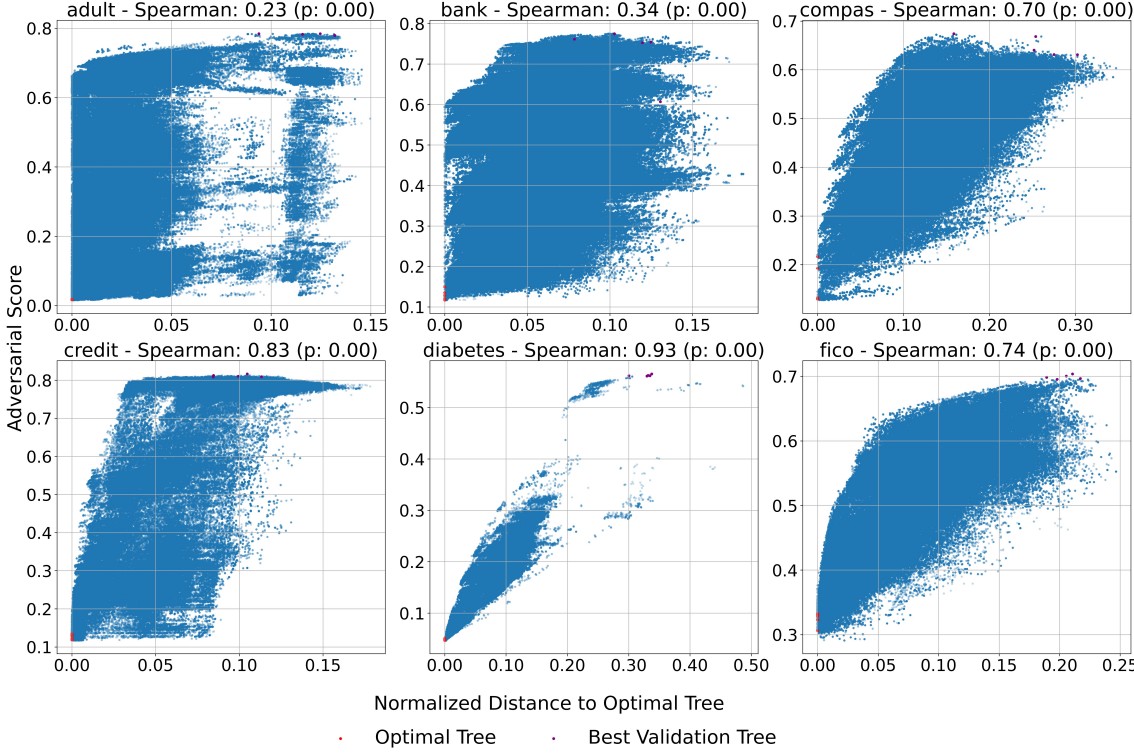

*Figure 7.* Adversarial score of trees in Rashomon set vs. their distance to optimal tree. Results are aggregated over five folds. The optimal trees (in red) are attacked. The most robust trees (in purple) are far from optimal tree.

Additionally, we show a plot of clean task accuracy vs adversarial accuracy in Figure 8. This figure demonstrates that all trees in the Rashomon set perform near-optimally, but they can have very different reactive robustness properties as discussed in our previous results.

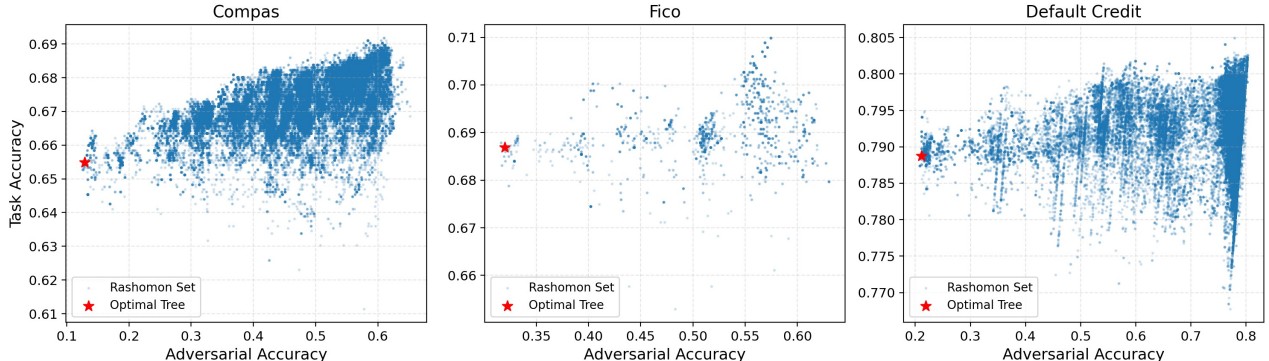

*Figure 8.* Adversarial accuracy vs task accuracy of trees in Rashomon set on different datasets on the validation dataset. Each blue point represents a tree in the Rashomon set. The adversarial examples are generated by targeting the optimal tree, which is represented by a red star.

### Diversity in the Rashomon Set Accelerates Information Leakage

**Collection and Setup:** We ran this experiment on 6 datasets. Following the setup in (Ferry et al., 2024), we binarized each dataset and subsampled 100 data points to form the training set. We ran TreeFARMS on these 100-sample datasets using $\epsilon = 0.02$ and depth_budget= 5. We tuned the regularization parameter instead of $\epsilon$ to control the size of the Rashomon set, since the set size is less sensitive to changes in the regularizer. Controlling the size is necessary because the DRAFT algorithm can only feasibly compute ensembles with a few hundred estimators (Ferry et al., 2024).

The exact values of the regularizer and the average number of trees are reported in Table 5. Once the Rashomon set is constructed, trees are sequentially selected from the Rashomon set and passed to DRAFT. We run DRAFT multiple times as more trees are added. Specifically, DRAFT is trained after each additional tree from 1 to 50, every 5 trees from 50 to 150, and again at 175 and 200 trees. In total, we consider up to 200 trees from the Rashomon set. We run this process five times with different random seeds for sampling data points. We consider two strategies to select trees. By default, the first tree selected is the optimal model. The *closest* strategy then iteratively selects the tree whose classification pattern has the smallest Hamming distance to that of the optimal tree. The *farthest* strategy greedily selects the tree whose classification pattern has the largest Hamming distance from those of the previously selected trees.

*Table 5.* Summary of parameters for the data reconstruction experiment and the average number of trees across five runs.

| Dataset | adult | bank | compas | credit | diabetes | fico |
|---|---|---|---|---|---|---|
| Regularization $\lambda$ | 0.01 | 0.013 | 0.01 | 0.0165 | 0.0125 | 0.02 |
| Average # Trees | 29428.6 | 12267.4 | 27432.8 | 6806.8 | 6418.4 | 7327.0 |

**Results:** Figure 9 shows the comparison of reconstruction error between different selection strategies. For all datasets, the "farthest" strategy (in purple) has lower reconstruction error, indicating greater information leakage. The "closest" strategy (in green) has better privacy, as it has higher reconstruction error. These results suggest that more diversity in the Rashomon set and disclosing more diverse models lead to increased privacy risk. The random baseline randomly guesses the feature values for each data point. It is a very conservative attack and usually results in high reconstruction error.

### Robustness-Privacy Tradeoffs under the Rashomon Set
**Collection and Setup:** We evaluate the robustness-privacy tradeoff directly in this experiment. First, we construct multiple groups of trees of varying sizes from a Rashomon set to represent different levels of diversity. Then, we assess each group's performance under both reconstruction and robustness attacks. To form these groups, we sort trees in the Rashomon set by Hamming distance of their classification patterns to the optimal tree, then select trees at evenly spaced intervals. For

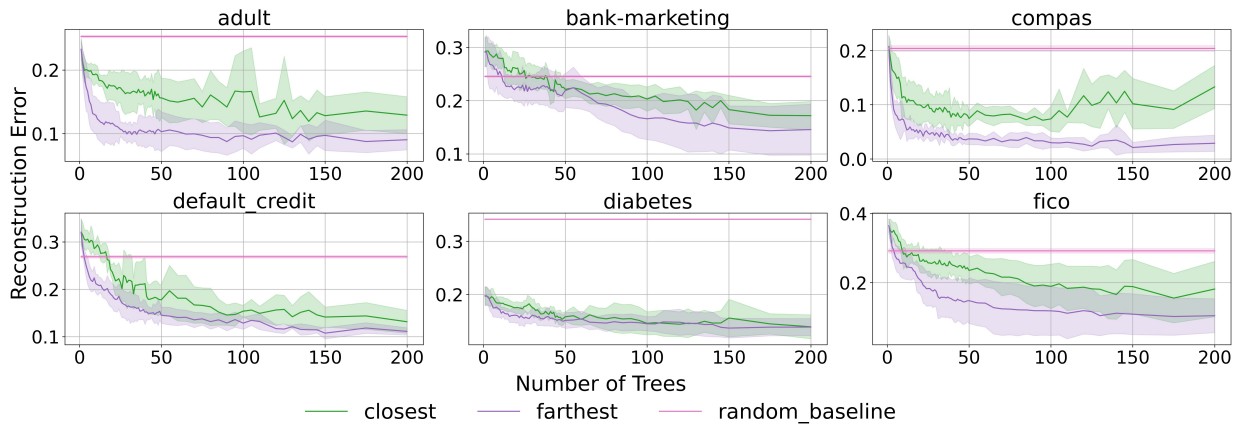

*Figure 9.* Comparison of reconstruction error between different selection strategies. The random baseline randomly guesses the feature values for each data point.

instance, given a Rashomon set of 1000 trees, an ensemble of 3 trees would include those ranked 1, 500, and 1000, while an ensemble of 100 trees would include every 10th tree in the sorted list. As in previous experiments, we use DRAFT to perform the reconstruction attack and report the reconstruction error for each group. For the robustness evaluation, we apply an adversarial attack targeting the optimal tree in the Rashomon set using the $\mathcal{S}_0$ set with $\eta = 1$, which allows a single binary feature flip per data point. This setup is used because DRAFT requires binary features. We then evaluate all trees within each group and record the adversarial accuracy of the best-performing tree in the group.

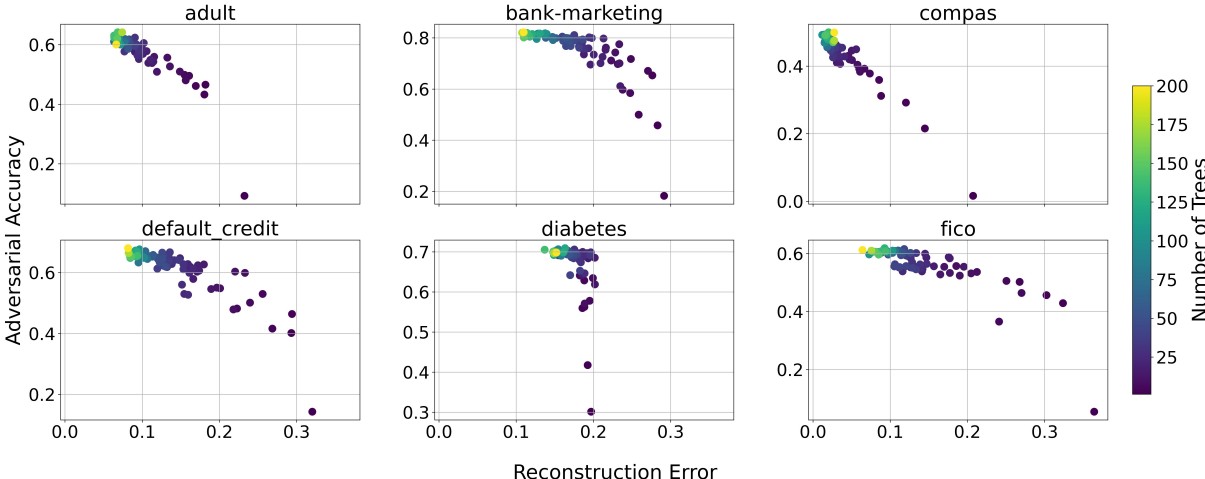

*Figure 10.* Reconstruction error vs. adversarial accuracy for ensembles constructed with different numbers of evenly sampled trees from the Rashomon set.

**Results:** Figure 10 shows the reconstruction error versus adversarial accuracy for ensembles constructed by selected trees. Each point represents an ensemble of a specific size, with color indicating the number of trees included. When only a limited number of models are released, the privacy is preserved but the models are not diverse enough to avoid an adversarial attack targeted on the optimal trees. As more trees are selected, robustness improves but at the cost of greater information leakage.

### D.4. Study for Theorem 5.5

We provide in this section empirical verification of the theoretical results of Theorem 5.5. To do this, we compute the Rashomon sets for 4 binarized datasets using TreeFARMS with regularization equal to $0.01$, Rashomon parameter $\epsilon = 0.03$, and depth_budget $= 5$. For each dataset and 24 values of $K$ uniformly ranging from $0.25\%$ to $6\%$ of the number of points of the dataset, we modify $K$ samples of the dataset. Specifically, the modification begins by using the $k$-means clustering

algorithm to compute 5 clusters of the dataset. We randomly select a cluster with over $K$ samples, and we uniformly and at random remove $K$ samples from that cluster. We then randomly select a different cluster, from which we randomly and with replacement select $K$ samples to duplicate. Lastly, we flip the label of each of the duplicate samples with probability $50\%$. The resulting dataset sees a targeted shift in both the feature distribution as well as the label distribution.

On the modified dataset, we compute the modified Rashomon set with the same value of regularization and depth_budget as before, as well as Rashomon parameter $\epsilon'$. We use two different values of $\epsilon'$, where $\epsilon' = \epsilon$ reuses the same Rashomon parameter as before, and $\epsilon' = \epsilon + \frac{2K}{n}$ uses the Rashomon parameter stated in Theorem 5.5. We lastly compute the percentage of models in the original Rashomon set that remain in the modified Rashomon set. We repeat this process on 5 modified datasets in total and average the results.

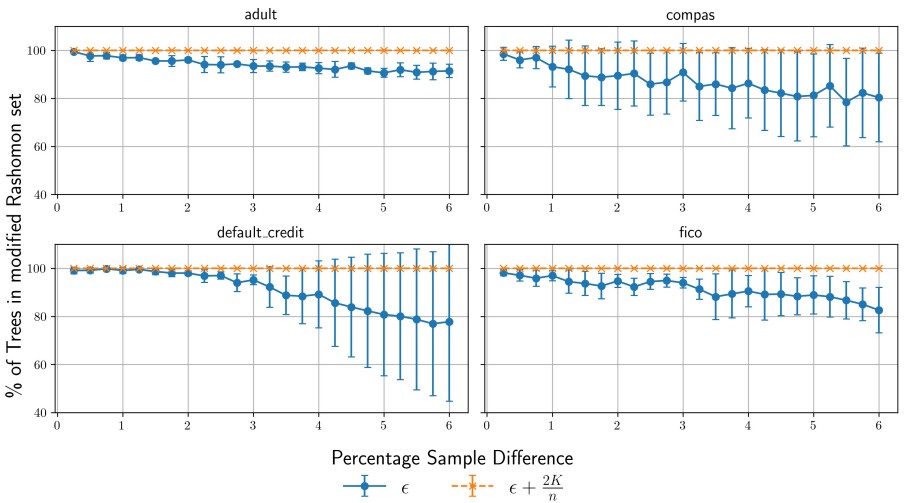

*Figure 11.* Percentage of trees in the Rashomon set that remain in the modified Rashomon set after modifying $K$ samples.

As we observe in Figure 11, we can see that every model in the original Rashomon set remains in the modified Rashomon set when $\epsilon' = \epsilon + \frac{2K}{n}$. This agrees with Theorem 5.5, which predicts that the original Rashomon set should be contained within the modified Rashomon set. When we fix $\epsilon' = \epsilon$, we instead see that some models exit the Rashomon set after we modify the dataset, with a greater change in the dataset resulting in a smaller overlap between the two Rashomon sets.

### D.5. Synthetic Study for Theorem 5.6

In Section 5.3, to support analytical conclusions of Theorem 5.6, we provided a simulation experiment where we measure the KL divergence between the synthetic distribution and the predicted distribution. The details of the simulation setup are described in this section.

We use five different data distributions for this experiment. Let $\mathbf{x} = (x_0, x_1, \ldots, x_{d-1}) \in \{0, 1\}^d$ be a binary feature vector of dimension $d$. We set $d = 4$ in this experiment. Let $P(Y = 1 \mid \mathbf{x})$ denote the true conditional probability of label $Y = 1$ given $\mathbf{x}$. Also, let the sum of features be $s = \sum_{i=0}^{d-1} x_i$. The five distributions are defined as follows:

- Distribution 1 (Parity): $P(Y = 1 \mid \mathbf{x}) = 0.1 + 0.8 \cdot \mathbb{1}[s = 0]$.

- Distribution 2: $P(Y = 1 \mid \mathbf{x}) = 0.15 + 0.7 \cdot \mathbb{1}[x_0 = 1 \wedge x_1 = 1]$.

- Distribution 3 (XOR): $P(Y = 1 \mid \mathbf{x}) = 0.2 + 0.6 \cdot \mathbb{1}[(x_0 \oplus x_1 = 1) \wedge (x_2 \oplus x_3 = 0)]$.

- Distribution 4 (Random): $P(Y = 1 \mid \mathbf{x}) = 0.3 + 0.4 \cdot r, r \sim \mathcal{U}(0, 1)$.

- Distribution 5: $P(Y = 1 \mid \mathbf{x}) = \begin{cases} 0.05 & \text{if } s \leq 1 \\ 0.95 & \text{if } s \geq 3 \\ 0.5 + 0.4(x_0 - 0.5) & \text{otherwise.} \end{cases}$

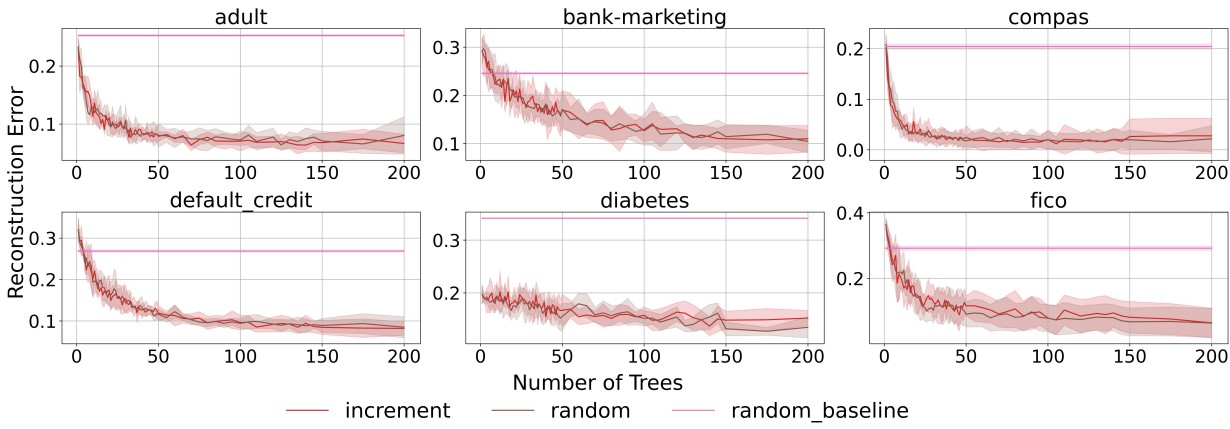

*Figure 12.* Comparison of reconstruction error between *increment* and *random* strategy. The random baseline guesses the feature values for each data point.

We set $d = 4$, the distribution $P(\mathbf{x})$ is uniform. For each of these five true distributions $P(Y = 1|\mathbf{x})$, we sample a dataset of 100 points 5 times. These train sets are then used to train the Rashomon set. The TreeFARMS configuration includes a regularization parameter of $0.001$ and a Rashomon bound multiplier of $0.03$.

For each training data, we consider ensembles of models from the Rashomon set. We start from one tree and increase counter $J$ that corresponds to the ensemble size until we reach the ensemble that consists of all trees in the Rashomon set. We estimate the expected KL divergence between the true distribution and an ensemble's average prediction. Our procedure is as follows for each $J$: (1) We sample an ensemble of size $J$ 20 times without replacement from the models in the Rashomon set. (2) For each ensemble and for every data point, we estimate the ensemble's average predicted probability of $P(Y = 1|\mathbf{x})$ by averaging predictions of models in the ensemble. (3) For each $\mathbf{x}$, we compute the pointwise KL divergence between the true conditional distribution $P_{\text{true}}(Y = 1|\mathbf{x})$ and the ensemble's predicted conditional distribution $P(Y = 1|\mathbf{x})$. (4) We compute the expected KL divergence by taking the empirical mean of pointwise KL divergences over 20 samples of the ensembles. (5) Finally, we report the expected KL divergence averaged over five datasets sampled from the same true distribution.

As we observe in Figure 5, the KL-divergences decrease as the number of trees in ensemble increases, verifying the results proved in Theorem 5.6.

## E. Additional Studies

### E.1. Analysis of Ensemble Construction Strategies

In Section 6 and Appendix D.3, we have introduced several strategies for constructing ensembles using trees from the Rashomon set. Here, we briefly summarize each strategy along with its motivation. We also introduce a few additional strategies that enable further interesting analyses.

**Random versus Evenly-spaced Sampling:** As mentioned in section 6.3, we introduce an *increment* sampling strategy, which selects trees from the Rashomon set at evenly spaced intervals based on their prediction patterns, depending on the size of the ensemble we want to construct. The goal is to evenly represent the Rashomon set in order to observe its default properties. We compare this strategy to the *random* sampling method, where trees are randomly selected to form the ensemble. For both methods, we include the optimal tree as the first selected tree by default. Figure 12 shows this comparison. We can observe that both strategies produce closely aligned error values and follow similar trends. Therefore, using the increment sampling can help us capture the variation in the Rashomon set.

**Closest, Farthest, and Evenly-spaced Sampling:** In section 6.2, we introduced the *closest* and *farthest* sampling strategies. The closest strategy selects the next tree that is most similar to the optimal tree in terms of prediction patterns. In contrast, the *farthest* strategy aims to maximize diversity by greedily selecting trees that have the highest average prediction distance from those already selected. We compare these strategies with the *increment* strategy.

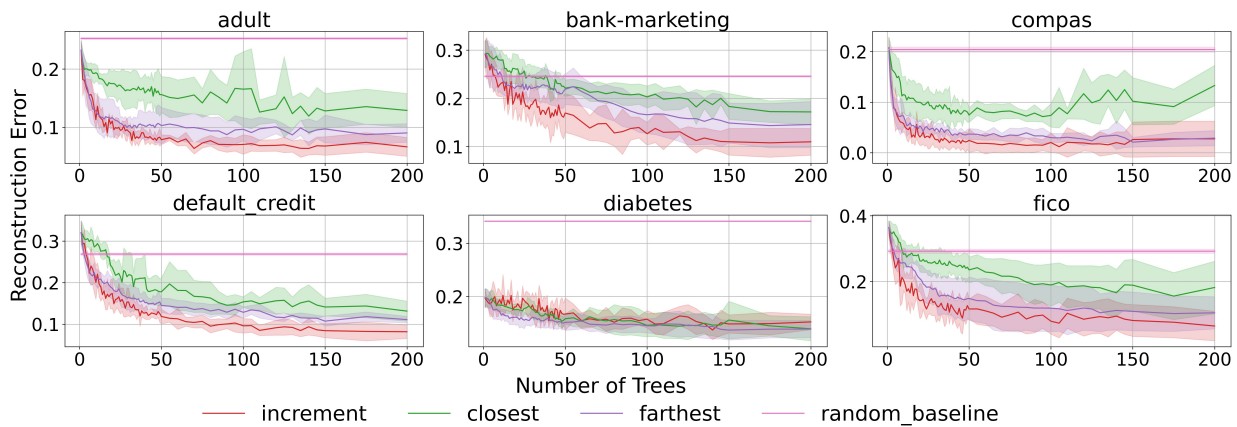

*Figure 13.* Comparison of reconstruction error between *increment*, *closest*, and *farthest* strategy. The random baseline guesses the feature values for each data point.

Figure 13 shows that the increment strategy leaks more information than the other two strategies, indicating that evenly spaced sampling may pose a higher privacy risk. This is not particularly surprising, as selecting only similar or highly dissimilar trees tends to concentrate on specific regions of the Rashomon set, whereas the increment strategy more effectively captures the full landscape. This finding also motivates future research into alternative definitions of diversity beyond prediction patterns.

**Sampling Strategies Based on Tree Sparsity:** In addition to diversity-based and evenly-spaced sampling strategies, we also explore other approaches based on sparsity. In decision trees, sparsity is usually measured by the number of leaves. We study two sparsity-based strategies: the *sparsest* strategy selects trees with the fewest leaves in ascending order, while the *densest* strategy selects trees with the most leaves in descending order.

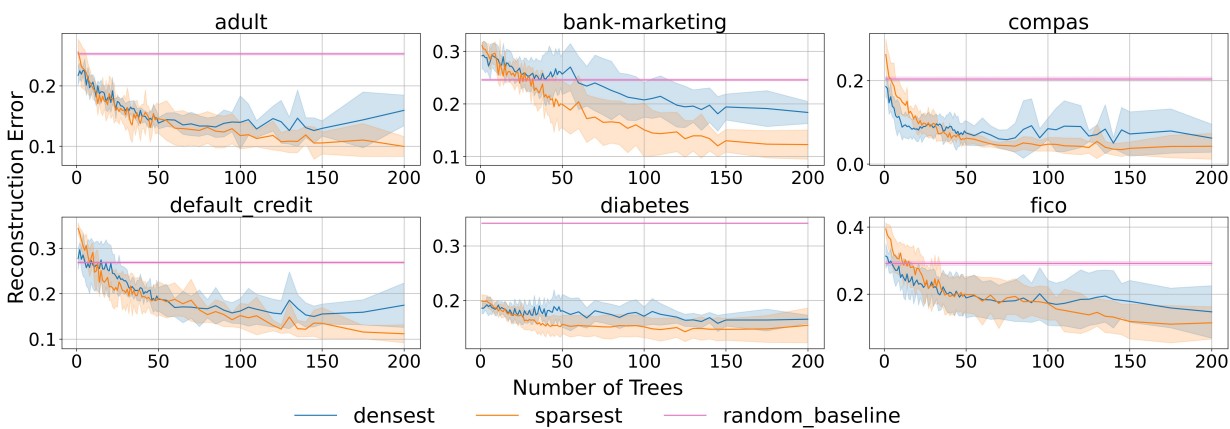

*Figure 14.* Comparison of the reconstruction error between *sparsest* and *densest* strategy. The random baseline guesses the feature values for each data point.

Theorem 4.1 proves that individual sparse models leak less information. This is reflected in Figure 14, where the sparsest tree (in orange) has higher reconstruction error than the densest tree (in blue) when the x-axis is one. However, as more trees are added, the orange curve drops below the blue curve. This may be because sparse trees, while individually less expressive, offer greater structural diversity and better generalization when aggregated. As a result, ensembles of sparse trees can more effectively cover the input space without overfitting, leading to lower reconstruction error compared to ensembles composed of dense trees, which may be more redundant or over-specialized. This observation suggests that sparsity may have a more complex impact on privacy leakage and motivates further study of how structural factors influence privacy under the Rashomon set setting.

**Comparing Robustness-Privacy Trade-offs for Different Sampling Strategies:** In Section 6 Figure 4, we empirically

observed the robustness-privacy trade-offs when the Rashomon set is present. In this figure, we used the incremental sampling procedure. Here, we verify that other sampling strategies lead to this trade-off as well. Experimentally, we use the same setup as in Figure 4.

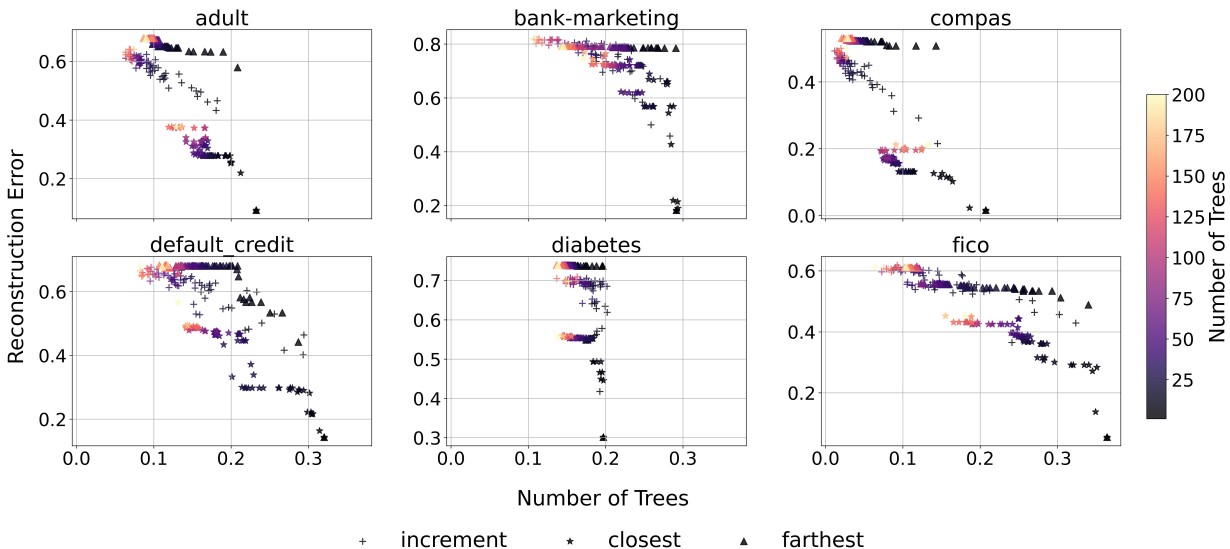

*Figure 15.* Reconstruction error vs. adversarial accuracy for ensembles constructed with *increment*, *farthest*, and *closest* strategy. Each strategy is plotted with different markers as shown in the legend.

In Figure 15 we plot the robustness-privacy trade-off for six datasets under different sampling strategies. First of all, we notice that the trade-off is preserved for all three strategies. However, the extent of it differs, most likely due to the diversity of the constructed Rashomon sets (for example, we expect the farthest strategy to produce less diverse sets as compared to incremental one). This difference in the extent of the trade-off between sampling strategies is especially evident in datasets like COMPAS and Adult.

Besides the trade-off, we also saw another interesting pattern in this figure. For some datasets like bank-marketing and diabetes, we observe that one can construct a Rashomon set that contains models that can achieve low privacy risk and high robustness (as indicated by the presence of points in the top right of the plots). Overall, our empirical findings show that this trade-off is an interesting phenomenon that is influenced by the diversity of the Rashomon set and can be a rich direction for future research.

### E.2. Single Tree is Vulnerable to Adversarial Attack

In Section 5.1 in Theorem 4.2, we discussed the inherent vulnerability of a single model, using rule lists, which is a one-sided decision tree. Here, we verify empirically this vulnerability for the Rashomon set of sparse decision trees, which include rule lists as well. Our setup is similar to the one in adversarial robustness experiments that we described above. For COMPAS, default-credit, and FICO, we used binarized datasets as described in Section 6.2. We did not perform subsampling for the Rashomon set. We divided the data into five folds for cross-validation. We trained TreeFARMS with regularization of $0.01$, depth budget of $4$, and the Rashomon adder $\epsilon$ set to $0.01$. We performed $\ell_1$ attack with $\eta = 1$, allowing a single binary flip to each row. We attacked each tree separately.

The results of this experiment are presented in Figure 16. There was only one tree in the credit dataset for which the attack was ineffective. For all other trees across the three datasets, we observed a steep drop in accuracy of at least 60% between the original empirical risk, $\hat{L}_S(tree)$, and the adversarial risk, $\hat{L}_{S'}(tree)$, where $tree$ is a model from the Rashomon set. Therefore, within the hypothesis space of sparse decision trees, there is an inherent vulnerability when each tree is attacked individually (similar to Theorem 4.2). As we show in Section 5.1, diverse Rashomon sets allow for model selection that is more resilient to the attack, provided the notion of diversity aligns with the type of attack.

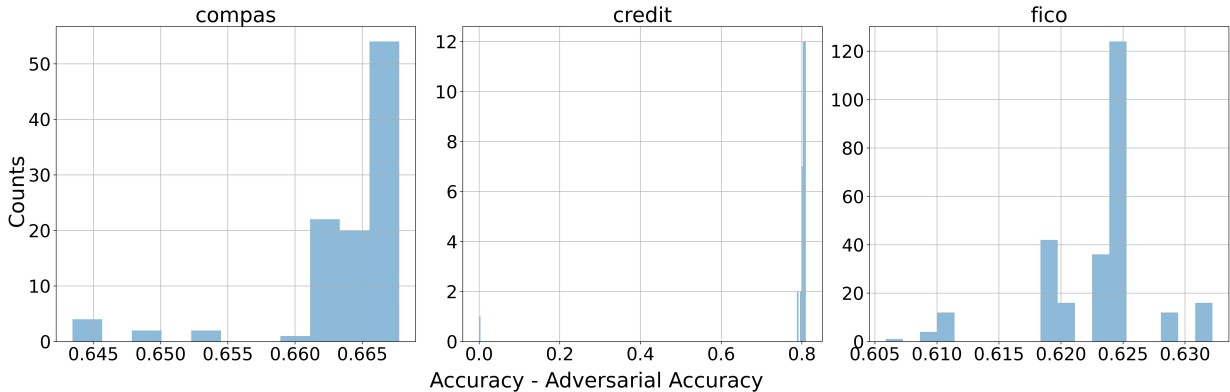

*Figure 16.* Accuracy gap ($\hat{L}_{S'}(f) - \hat{L}_S(f)$) under $\ell_1$ attack on the individual trees of the Rashomon set of sparse decision trees. The results presented in histogram are averaged over five folds.

### E.3. Trees are Robust to Simultaneous Attacks

In Section 6, we evaluate reactive robustness by showing that attacks on the optimal model transfer poorly to diversely chosen models in the Rashomon set. However, in a real-world setting where models from the Rashomon set are released gradually over time, an adversary may create attacks that target multiple released models simultaneously, resulting in attacks that may work against many models in the Rashomon set at the same time. In this section, we empirically investigate this for decision trees and demonstrate that the Rashomon set is resilient to simultaneous multi-model attacks.

We reuse the experimental setup in Section 6.2. For each of 6 datasets, we subsample 100 points to form the Rashomon set of decision trees. We sequentially select trees from the Rashomon set using the *farthest* strategy. As each tree is added, we form an adversarial dataset by finding adversarial samples that attack each already-chosen tree simultaneously via a naive exponential-time search over all joint decision regions of the trees. Due to the high complexity of this procedure, we only select up to 10 trees. To our knowledge, there is no known efficient strategy for attacking multiple trees simultaneously, which is fundamentally different from attacking an ensemble. For comparison, we also consider a single-model attack done by performing an adversarial attack on the optimal tree using the algorithm proposed in (Kantchelian et al., 2016).

For each adversarial dataset, we evaluate the adversarial accuracy of each selected tree. We plot the best, mean, and worst adversarial accuracy over each tree in Figure 17. We observe that, for the single-model attack, the best tree has higher adversarial accuracy, and the worst tree (the one that's targeted) has lower accuracy than their counterparts under the multi-model attack. This suggests that, while the single-model attack is stronger than the multi-model attack on the optimal tree, it's overall weaker across the entire Rashomon set. The multi-model attack is instead stronger overall, but could potentially be weaker on specific trees. This is as we expect.

As we add more trees, we see that the adversarial accuracy under the multi-model attack increases on each of the best, worst, and mean trees. This is likely because, as the multi-model attack targets more trees, it's more difficult to find an attack that works well on all of them due to the diversity of their predictions, so the attack strength on any individual tree decreases. This suggests that the diversity of the Rashomon set forces adversaries to choose between creating strong attacks that transfer poorly, or creating weak attacks that work uniformly well over the Rashomon set. We lastly note that the computational difficulty associated with attacking multiple models simultaneously further reduces the efficacy of potential adversarial attacks that use multiple released models.

### E.4. Model Diversity is Encouraged Under Adversarial Attacks

In Theorem 5.1, we theoretically show that for linear models under margin-based losses, the transferability of an adversarial attack is directly related to the similarity between the models being attacked. We furthermore give a closed-form expression for the change in loss under the adversarial attack for the exponential loss. In this section, we empirically verify these results for the hypothesis space of neural networks. Through these experiments, we suggest that the trends established by Theorem 5.1 hold more generally.

**Experimental Setup:** We work with the Iris and Seeds binary classification datasets from the UCI Machine Learning

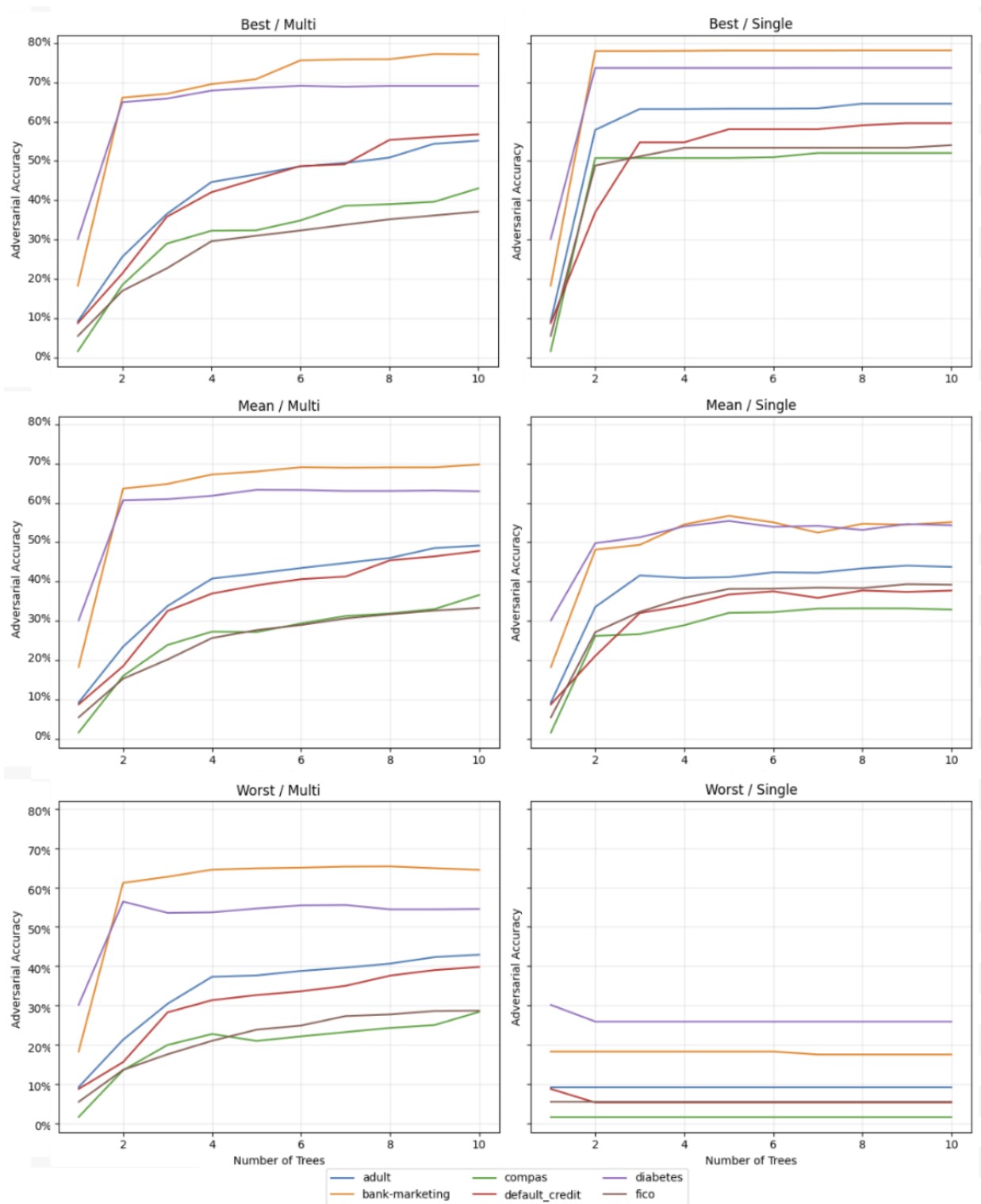

*Figure 17.* Adversarial accuracy statistics of subsets of the Rashomon set under multi-model and single-model attacks. The results are averaged over five random seeds.

Repository (Dua & Graff, 2017) as well as the larger COMPAS recidivism dataset, each of which we normalize so that the features are in the range $[0, 1]$. We use cross-entropy loss and we first compute the optimal model using stochastic gradient descent. We use a basic feed-forward neural network structure. The network parameters for each dataset are listed in Table 6. Since the full Rashomon set is not computationally feasible to compute, we estimate the Rashomon set using the Adversarial Weight Perturbation (AWP) algorithm (Hsu & Calmon, 2022). We set the multiplicative Rashomon parameter to be $0.8$ (i.e. $\epsilon = 0.8\hat{L}(\hat{f})$) for the UCI datasets and $0.5$ for the COMPAS dataset. For varying levels of attack strength $\eta$, we adversarially attack the optimal model using the FGSM attack (Goodfellow et al., 2015), which constrains the $\ell_\infty$ norm of the perturbation. Using the perturbed data, we construct an adversarial dataset on which we find the best-performing model in the Rashomon set. We then measure the similarity between this model and the original optimal model using their pattern similarity–the percentage of matching predictions on the original dataset.

*Table 6.* Summary of parameters for the hypothesis space of feed-forward neural networks.

| Dataset | Iris | Seeds | COMPAS |
|---|---|---|---|
| # Layers | 2 | 3 | 4 |
| # Hidden Dimension | 8 | 10 | 20 |

**Results:** For each dataset, we plot the similarity between the optimal models on the original dataset and the adversarial dataset as $\eta$ is varied. We furthermore plot the loss of the two models on the adversarial dataset as $\eta$ changes. We visualize this in Figures 18 and 19. We observe that, as the strength of the adversarial attack is increased, our procedure selects models that are increasingly dissimilar to the original optimal model. This suggests that adversarial attacks transfer better to more similar models, causing more similar models to suffer higher drops in transferred adversarial loss compared to less similar models. This matches what we would expect from our theoretical results for Theorem 5.1, supporting the usage of diverse models in the Rashomon set to defend against adversarial attacks.

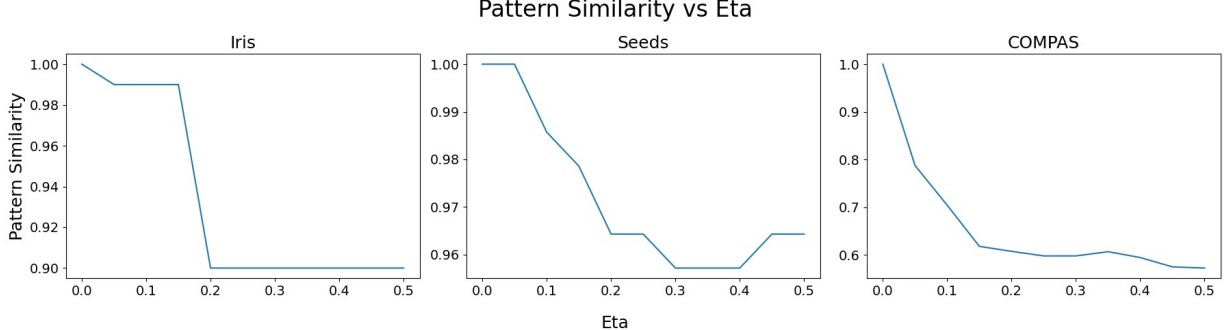

*Figure 18.* Prediction similarity under $\ell_\infty$ attack.

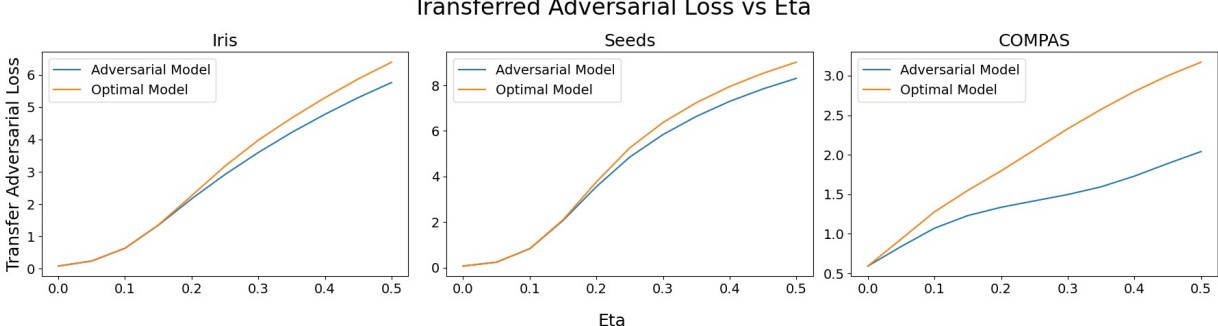

*Figure 19.* Optimal adversarial loss under $\ell_\infty$ attack.

## E.5. Robustness-Privacy Tradeoffs with Neural Networks

To see if our analysis on interpretable models may extend to deep learning architectures, we apply a similar setup as described in the previous section to perform similar evaluations. We use Iris, Seeds, and Wine datasets from UCI and have 2 layers with 16 hidden dimensions for all 3 of them. We also use a $\epsilon = 0.3\hat{L}(\hat{f})$ for the multiplicative Rashomon parameter.

For adversarial robustness, we measure the best adversarial accuracy achievable from a pool of models under attack strengths $\eta = 0.5$ using FGSM. The experiment simulates a scenario where, given a large pool, one can select the most robust model against adversarial examples. The result is shown in Figure 20. We observe that increasing the pool size generally allows us to find a robust model where adversarial examples cannot transfer, but the degree of improvement varies by dataset and attack strength. For example, in the Seeds dataset, even a pool size of just 2 models gives drastic improvements in the system's reactive robustness.

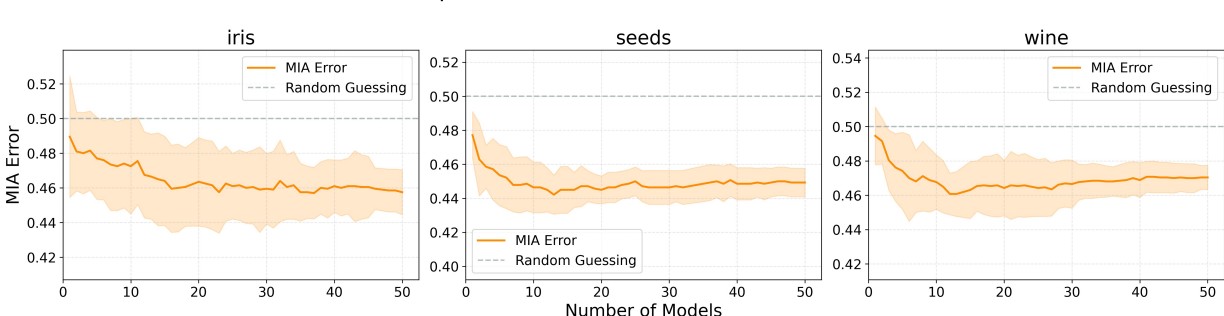

*Figure 20.* Best adversarial robustness vs number of models considered. For each size, 20 trials with different random seeds are run to calculate standard deviation. The horizontal line shows the average adversarial accuracy of reference model.

For privacy, we assess risk using the Yeom loss-threshold MIA (Yeom et al., 2018), which attempts to distinguish training set members from non-members based on model loss. For each ensemble size, we report the MIA error (1 - MIA accuracy), where higher values indicate better privacy. This is shown in Figure 21. Our results show that privacy risk can increase with ensemble size, but the relationship is not always monotonic and depends on dataset characteristics and model diversity.

Figure 22 combines these results in joint scatter plots (MIA error vs. best adversarial accuracy, colored by ensemble size) and visualizes the inherent trade-off between robustness and privacy. The experiments reveal that while larger ensembles can enhance robustness, they may also expose models to greater privacy leakage. This result is consistent with our findings in sparse decision tree hypothesis space.

**Applying the reactive robustness framework to the Rashomon set of Concept Bottleneck Models**   To demonstrate that our analysis generalizes beyond tabular domains, we extend our study on reactive robustness to a subset of the Rashomon set of Concept Bottleneck Models (RashomonCBMs) (Feng et al., 2025) for image classification. RashomonCBM uses a ViT-small backbone pretrained on ImageNet-21k with a concept bottleneck layer consisting of pre-defined linear concept

*Figure 21.* Membership inference attack error vs number of models. 20 trials with different random seeds are run to generate standard deviations. A horizontal reference line indicates the random guessing base threshold.

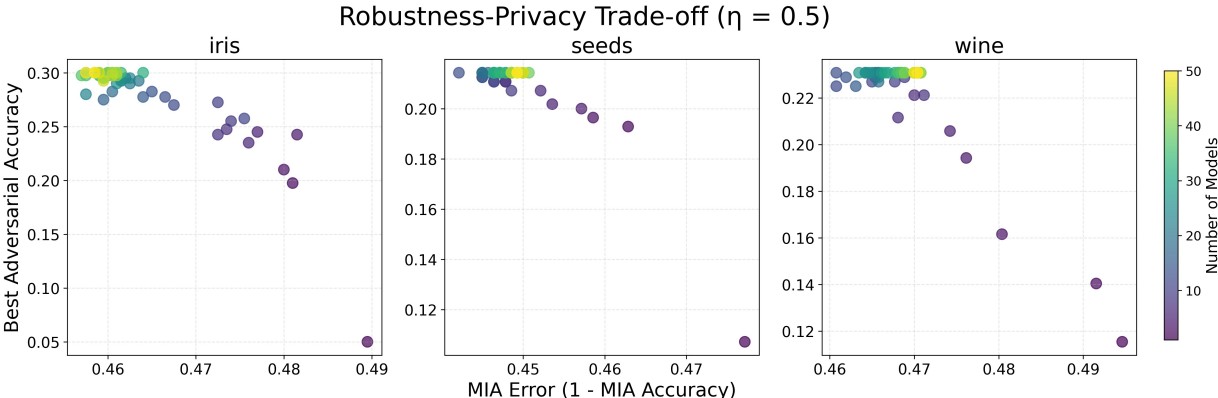

*Figure 22.* Membership inference attack error vs. best adversarial accuracy for ensembles constructed with different numbers of randomly sampled models from the Rashomon set.

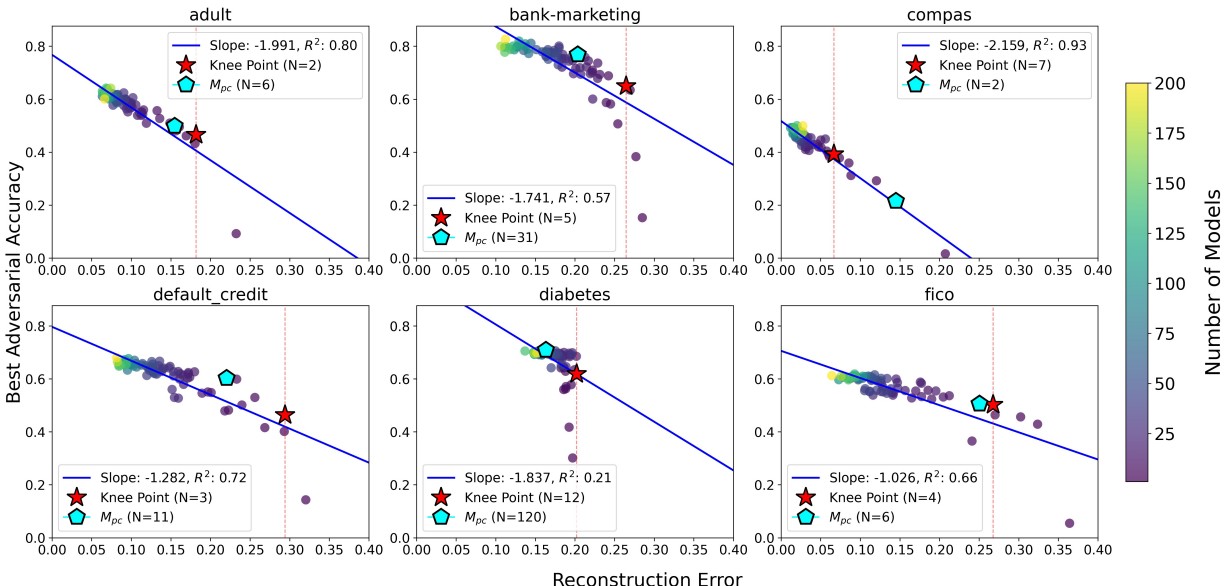

*Figure 23.* Reconstruction error vs. adversarial accuracy for ensembles constructed with increment selection across six datasets. The knee point is plotted as a red star and privacy-constrained point $M_{pc}$ as a cyan pentagon. $\delta$ value is 0.35.

heads, followed by linear classifiers mapping concepts to image classes. RashomonCBM trains parallel adapters that share the same backbone yielding a subset of the Rashomon set whose models rely on different reasoning logic. We restrict our analysis to reactive robustness, because the frozen backbone is shared across all models and the information each model leaks about the training data varies little with model selection. Therefore, membership inference vulnerability is more invariant within this Rashomon set.

We evaluate reactive robustness via attack transferability on the AwA2 dataset (Xian et al., 2018). Using the ten models from the original work, we attack each model individually and evaluate the resulting adversarial examples on all ten. For each source model, we use a projected gradient descent (PGD) attack with $L_\infty$ norm, $\epsilon = 8/255$, step size $2/255$, and 10 steps, targeting the task loss, the concept loss, or both jointly. The generated adversarial samples are used to evaluate on every model in the Rashomon subset.

Our results show that adversarial examples generated against a source model substantially degrade its own accuracy but transfer poorly to other members of the Rashomon set. In Table 7, the source row shows the adversarial accuracy of the model being targeted, averaged across the ten choices of source model, and the target row shows the average accuracy of the other 9 models in the Rashomon set under the same attack. The Recovery row is the gap between the two, i.e., the accuracy

regained by switching from the attacked model to a non-targeted member of the set. We can see for both concept and task accuracy, the non-targeted models are more resilient, confirming that reactive robustness holds in the image domain as well as the tabular setting.

*Table 7.* Attack transferability across Rashomon CBMs on AwA2. PGD examples generated on the source model transfer poorly to other models in the Rashomon set. Results averaged over the ten source models.

|  | **Task Accuracy (%)** | **Concept Accuracy (%)** |
|---|---|---|
| Clean (all models) | $97.27 \pm 0.19$ | $99.03 \pm 0.03$ |
| Source (white-box) | $43.95 \pm 1.14$ | $85.62 \pm 0.39$ |
| Target (transfer) | $61.68 \pm 0.90$ | $90.47 \pm 0.34$ |
| **Recovery** | **+17.73** | **+4.85** |

### E.6. Additional Details on Trade-off Operationalization

In this section, we provide the mathematical formulation for the "privacy-first" selection rule discussed in the main text. Acknowledging that there is no universal criterion for optimal selection along a Pareto frontier, we propose a procedure that serves as a practical starting point for determining the permissible model count $N$ subject to privacy constraints.

### E.7. Quantifying Trade-off Efficiency

We first quantify the global efficiency of the Rashomon set using a linear approximation of the Pareto frontier. Let $r$ denote the best adversarial accuracy and $p$ denote the privacy metric (e.g., reconstruction error) for a set size $N$. We model the relationship as $r \approx \beta p + c$, where $\beta$ is a slope, magnitude of which serves as the efficiency coefficient.

### E.8. The Hybrid Selection Algorithm

To determine the optimal set size $M_{final}$, we impose two distinct upper bounds: a privacy-based constraint ($M_{pc}$) and an efficiency-based constraint ($M_{knee}$).

For the privacy constraints, we define a hard privacy floor relative to the dataset's intrinsic difficulty. Let $E_{base}$ be the reconstruction error of a dataset-specific baseline. We permit the reconstruction error $E(N)$ to degrade only within a tolerance $\delta$ (we use 35% relative deviation, but any other relative or absolute rule can be applied) of this baseline. Then:

$$M_{pc} = \max\{N \in \mathbb{N} \mid E(N) \geq (1 - \delta) \cdot E_{base}\}.$$

This constraint ensures that the adversary's advantage does not exceed a statistically significant threshold, preserving plausible deniability regardless of potential robustness gains.

While we adopt a 35% tolerance threshold ($\delta = 0.35$) for this analysis to accommodate the discrete granularity of model sets, this value is intended to be illustrative rather than prescriptive; in practice, $\delta$ serves as a tunable safety parameter that stakeholders should calibrate based on their specific domain-governance requirements and risk tolerance.

For the efficiency constraint ($M_{knee}$), to prevent diminishing returns, we identify the topological "knee" or "elbow" of the trade-off curve using the Kneedle algorithm. This identifies the point of maximum curvature, $M_{knee}$, where the marginal gain in robustness begins to saturate relative to the marginal cost in privacy.

For the final decision, the operational set size is defined as the conservative intersection of these two bounds:

$$M_{final} = \min(M_{pc}, M_{knee}).$$

Applying this formalism to the results in Figure 23 reveals three distinct operational regimes driven by the data distribution. The first is an efficiency-limited regime, characteristic of datasets like *Bank Marketing* and *Diabetes*, where the reconstruction task is intrinsically difficult (high $E_{base}$). This difficulty creates a large "safe zone" for model aggregation; for instance, in *Diabetes*, the safety constraint permits an expansive set size of $M_{pc} = 120$. However, the efficiency constraint detects saturation much earlier at $M_{knee} = 12$, dictating a final decision of $M_{final} = 12$. In this scenario, the knee point is the

active constraint, preventing the unnecessary deployment of over 100 additional models that would consume privacy budget for negligible performance improvement.

In contrast, the second regime is safety-limited, observed in vulnerable datasets like *COMPAS* where the data structure allows for rapid reconstruction. Here, the error curve collapses quickly, triggering the safety constraint at $M_{pc} = 2$, well before the efficiency knee at $M_{knee} = 7$. Consequently, the decision is bound to $M_{final} = 2$, demonstrating how the ethical mandate to limit leakage strictly overrides the potential robustness gains offered by the knee point.

Finally, we observe a regime of convergent constraints in datasets like *FICO* and *Adult*, where the safety and efficiency bounds nearly align. For example, in *FICO*, the safety limit ($M_{pc} = 6$) and knee point ($M_{knee} = 4$) are adjacent, suggesting that for these datasets, the natural saturation of robustness roughly coincides with the safe limits of data disclosure.

