# OpenReview forum: "The Double-Edged Nature of the Rashomon Set for Trustworthy Machine Learning"
_ICML.cc/2026/Conference — ICML 2026 spotlight_

### Official Review · Reviewer_2axa · 2026-02-14

**Soundness:** 3
**Presentation:** 3
**Significance:** 3
**Originality:** 2
**Overall Recommendation:** 5
**Confidence:** 4

**Summary:**

Rashomon Sets is a way to construct multiple diverse yet near-optimal models. The paper focuses on studying the effect of Rashomon Sets on adversarial robustness and privacy, hinting at a tradeoff between these two desiderata. Theoretical and empirical evidence are provided to support the claim.

**Compliance With Llm Reviewing Policy:**

Affirmed.

**Final Justification:**

The authors have fully addressed my concerns.

**Key Questions For Authors:**

## Questions:
- Why not consider [1] for finding Rashomon Sets for Concept-based models in empirical validations?

[1] Feng, Shihan, et al. "Many Ways to be Right: Rashomon Sets for Concept-Based Neural Networks." arXiv preprint arXiv:2511.19636 (2025).

**Limitations:**

yes

**Strengths And Weaknesses:**

## Strength:
1. The question of robustness and privacy of the Rashomon Set is an important question and the overall approach to answer this question is well-formulated.
2. The text is well-written.
## Weaknesses:
1. Attack model: If we assume that the adversary can access multiple models for reconstruction, they can also have access to multiple models for adversarial attacks. As the adversary attacks more models, the set of remaining models that are different from the attacked model is smaller. As a result, finding a robust model in the Rashomon set is harder. Without considering this, I find the current analysis of adversarial robustness of the Rashomon set to be insufficient.
2. Lack of standard accuracy in adversarial attacks: It is common knowledge that there is a standard-robust accuracy tradeoffs and that is not mentioned at all, at least in the main text. This is particularly needed for this paper because $L_{S'}(f) \leq L_{S'}(\hat{f}) + H(f, \hat{f})$ only tell us about an upper bound for the adversarial loss. What I would guess is the clean loss is lower bounded by the clean loss of the best model *minus* the slack $\epsilon$ of the Rashomon set, but there is an inherent tradeoff here: a larger $\epsilon$ is needed for a more diverse set, but this leads a lower value of the lower bound for the clean loss. Maybe I have laid out the theoretical analysis here, but I don't see this being analyzed in the current experiments.
3. Lack of relevant empirical evidence: I agree that decision trees and linear models are fine for theoretical analysis, but empirical analysis should go beyond these classes of models. As an example, common interpretable-by-design models, such as Concept Bottleneck Models [2] or Sparse Coding in a concept space [3], are essential.

[2] Koh, Pang Wei, et al. "Concept bottleneck models." International conference on machine learning. PMLR, 2020.

[3] Chattopadhyay, Aditya, Ryan Pilgrim, and Rene Vidal. "Information maximization perspective of orthogonal matching pursuit with applications to explainable ai." Advances in Neural Information Processing Systems 36 (2023): 2956-2990.

---

> ### Author Rebuttal · Authors · 2026-03-31
>
> Thank you very much for your review and feedback!
>
> **W1. Attack model:**
>
> We agree that it is more difficult to defend against multi-model attacks on the Rashomon set, though it is feasible. The practitioner can also tune the Rashomon parameter to find the right balance of model diversity and base model performance to make it easier to find robust models. To extend our claims to this setting, we modified the experiment in Section 6.3 to simulate multi-model attacks. For $N=1, \ldots, 10$, we sample $N$ decision trees from the Rashomon set using the “farthest” strategy. We form an adversarial dataset by finding adversarial samples that attack each model simultaneously via a naive exponential-time search over all joint decision regions of the models (we chose this approach as there is no known efficient strategy for attacking multiple equally-good models simultaneously, which is fundamentally different from attacking an ensemble). We then compute the highest adversarial accuracy among the models sampled, which we depict in the figure under this anonymized url: https://ibb.co/0yPHCrvc.
>
> We observe that the adversarial accuracy for the multi-model attack significantly increases with the number of decision trees. This indicates that, **despite the attack quality improving with $N$, the diversity within the Rashomon set ensures that adversarial accuracy continues to increase, indicating that robust models remain available even under multi-model attack.** The multi-model attack degrades adversarial accuracy compared to the single-model baseline (Figure 9), but the gap remains reasonable across most datasets. This is expected since the multi-model attack is stronger than the single-model attack.
>
> Furthermore, the time complexity of the multi-model attack scales exponentially with the number of trees. This exponential cost of the joint attack is itself a benefit of reactive robustness: since the adversary must consider multiple models simultaneously, performing such attacks becomes computationally expensive as N grows, raising the practical cost of sustained adversarial attacks.
>
> **W2. Lack of standard accuracy in adversarial attacks**
>
> This concern is addressed by the Rashomon set definition itself. Every model we consider satisfies $obj(f) \leq obj(\hat{f}) + \epsilon$ by definition, which directly bounds the clean loss of any selected model to within $\epsilon$ of the optimal loss. With $\epsilon$ set to 1-3% in our experiments (Table 3), no model in the Rashomon set can sacrifice meaningful clean accuracy and this is guaranteed by construction, not assumed. In our setting, diversity is constrained to models that are already near-optimal on clean data, which is the advantage of the Rashomon set framing that handles the robustness-accuracy trade-off as well.
>
> To make this more explicit, we will add a clarification in Section 5.1, and include a plot of clean vs. adversarial accuracy in the appendix. Figure 17 in the current paper already shows that the best adversarial model in the Rashomon set maintains competitive clean accuracy, further reinforcing this point.
>
> **W3+Q1: Lack of relevant empirical evidence.**
>
> We agree that extending empirical analysis to broader model classes is valuable. The paper already includes empirical evidence for multi-layer perceptrons in Appendix D.3-D.4 (Figures 16-20), showing that the robustness-privacy tradeoff is not limited to sparse decision trees.
>
> Following the reviewer's suggestion, we further evaluated reactive robustness on the Rashomon slice (a diverse subset of the Rashomon set) of Concept Bottleneck Models (CBMs) constructed using the framework of Feng et al. (2025) on the AwA2 dataset. Each of the ten models is used as an attack source and evaluated on all others, giving 100 source-target evaluations (10 self, 90 cross-model) on 1280 test samples using projected gradient descent attacks. **The key finding is that self-attacks are consistently stronger than transferred attacks: task adversarial accuracy is 43.95% for self-attacks versus 61.68% under transfer (+17.73%), stable across all ten sources (17.73 ± 0.96%).** Adversarial accuracy on concept predictions shows the same pattern with smaller magnitude (85.63% self vs 90.48% transfer, +4.85%). This confirms that attack transferability is lower between diverse models, which is consistent with Section 5.1.
>
> It is computationally expensive and unclear how we can perform meaningful privacy attacks for CBMs. For tabular data and NNs we use standard reconstruction attacks and Figure 4 (right) in the main text shows a full robustness-privacy tradeoff analysis.
>
> Thank you again for helping us improve our paper! We are happy to discuss any points during the discussion period.

---

> > ### Author Rebuttal · Reviewer_2axa · 2026-04-01
> >
> > (W1) I appreciate this experiment for clarifying my concern. To make sure that the attack quality actually increases, can I also get the adversarial accuracy of the worst/average tree? It could be the case that finding the adversarial attack is exponentially hard so we are just not finding a good attack. I do agree that if no efficient attack exists (which is the subject for another paper), requiring the adversary to exponentially search seems like a big plus, but that is a big if.

---

> > > ### Author Response · Authors · 2026-04-03
> > >
> > > We thank the reviewer for this helpful follow-up. In our added experiment, the multi-model attack is computed by brute-force enumeration over the joint decision regions of the selected trees. Therefore, the attack is exact. It is not computed by a heuristic optimizer. However, because of this exact brute-force nature, it has exponential runtime, so we could meaningfully perform it for small N (in our case N <= 10).
> > >
> > > Following suggestion, we report the best, mean, and worst adversarial accuracy of any of the attacked trees under the multi-model attack. Furthermore, for comparison, we plot the same metrics for single model attack on the optimal model described in detail in section C.3 corresponding to Figure 9. As in C.3., we investigate how this single-model attack transfers to other models in the Rashomon set as chosen by the “farthest” procedure.
> > >
> > > Anonymized links:
> > >
> > > Best vs Worst: https://ibb.co/kgMg5t1r
> > >
> > > Best vs Mean: https://ibb.co/wbYcG6c
> > >
> > > The figures match generally what we would expect. The single-model attack has greater variance in the adversarial accuracy of each of the trees. That is, the best tree has higher accuracy and the worst tree (the one that’s targeted) has lower accuracy than their counterparts under the multi-model attack. This suggests that, while the single-model attack is stronger than the multi-model attack on the optimal tree, it’s overall weaker across the entire Rashomon set. The multi-model attack is instead stronger overall, but could potentially be weaker on specific trees.
> > >
> > > Additionally, we see that the adversarial accuracy under the multi-model attack increases as we add more trees, on each of the best, worst, and mean trees. This is because, as the multi-model attack targets more trees, it’s more difficult to find an attack that works well on all of them due to the diversity of their predictions, so the attack strength on any individual tree decreases (please recall that this attack is exact). This suggests that the diversity of the Rashomon set forces adversaries to choose between creating strong attacks that transfer poorly, or creating weak attacks that work uniformly well over the Rashomon set. We will add these nuances to the revision.
> > >
> > > We note that **the fact that the adversarial accuracy increases even on the worst tree indicates that the attack quality is improving since the attack is considering a broader range of models in the Rashomon set**.
> > >
> > > In general, we agree that the exponential cost of attacking multiple models simultaneously is itself practically relevant: even when such attacks exist, requiring exact search over joint decision regions already makes the attacker’s problem substantially harder. It is indeed an interesting future research area to study such attacks and methods in more detail.

---

### Official Review · Reviewer_xRwu · 2026-03-12

**Soundness:** 3
**Presentation:** 3
**Significance:** 2
**Originality:** 3
**Overall Recommendation:** 3
**Confidence:** 3

**Summary:**

This paper proposes to explore the diversity of rashomon set (of models having nearly identical accuracy on a given task) from the perspective of robustness (to adversarial examples or covariate shift) and privacy. To that end, it provides some theoretical analysis and experimental assessment of the various properties of rashomon sets from a trustworthy ml perspective.

**Compliance With Llm Reviewing Policy:**

Affirmed.

**Final Justification:**

see the rebuttal

**Key Questions For Authors:**

-can you clarify contribution with respect to "The Rashomon Set Has It All: Analyzing Trustworthiness of Trees under Multiplicity", Neurips 2025 ?

-can you clarify the reactive robustness scheme ?  and the "natural policy object ?

-how does your theoretical analysis converges into outlining this double-edged nature ?

**Limitations:**

yes

**Strengths And Weaknesses:**

This paper addresses the important problem of trustworthy ML and provides a clear presentation of interesting results. Its objective of combining theoretical and empirical evidence is noble.

Moreover, it proposes an ambitious paradigm shift in which model providers would provide rashomon sets rather than individual models to regulatory inspection, or rely on rashomon sets to provide adversarial robustness using reactive robustness

On the downsides, research on rashomon sets and other predictive multiplicity has been quite a covered active research area ("Systemizing Multiplicity: The Curious Case of Arbitrariness in Machine Learning" provides an interesting overview) and the specific angles tackled in this approach feel quite anecdotic compared to the body of results already available.

The envisioned "reactive robustness" and rashomon regulatory release are intriguing. Unfortunately, they are never properly described. For instance in the reactive robustness proposal what is the attack model ? How are models rotated ? A discussion on the cost of such reactive robustness is missing too, I believe it is subjected to classical random forests vs tree discussion, e.g. memory size, loss in interpretability. Also, while rashomon sets of decision trees can be found using treefarms, how would this approach translate to larger, more complex model architectures ? Would providers still rotate larger neural networks ? I would have enjoyed a bit more details on the cost/benefits ratio of such reactive robustness, especially since "real attackers don't compute gradients (..)" (Apruzzese et al.,SATML'23).

Several theory results each rely on disconnected specific hypotheses (eg 4.1 on binary --balanced,complete-- decision trees, 4.2 rule lists, 5.1 linear models, 5.2 with exp loss), experiments are on trees. As a result, it is difficult to construct a coherent picture of the described landscape.
Moreover, the formulation of many of these results appear simplisitic, for instance section 5.3 could be clearer in its negative formulation: "larger rashomon sets cannot leak less information" where it appears obvious. Theorems 5.4 and 5.5 feel like "everything is equivalent for a large enough rashomon radius". Many properties (eg privacy, adversarial robustness) are indirectly manipulated (e.g. KL divergence with dataset instead of differential privacy bound). While I don't want to downplay the papers contribution, I believe they would benefit from a broader overview that would connect all these isolated results: how does the mentioned tradeoff emerges from these results ?

The experimental part is interesting but seems to borrow a lot from "The Rashomon Set Has It All: Analyzing Trustworthiness of Trees under Multiplicity", Neurips 2025, which is not cited. A clarification of the novelty here would help. I did not get the rationale for an "even sampling" (lines 421+)

---

> ### Author Rebuttal · Authors · 2026-03-31
>
> Thank you very much for your review and feedback!
>
> **W4+Q1 Novelty relative to The Rashomon Set Has It All**
>
> We will cite and discuss this work in the revision. Hsu et al (2025) is a *benchmark paper*: it asks whether one can find an *individual* model in the Rashomon set that is robust, fair, or stable, evaluating each property separately per model. Their conclusion lists “how should we quantify diversity to predict trust gains?” as future work. Our paper studies this question at the *set level*. We show that diversity within the Rashomon set simultaneously enables robustness (diverse models fail differently under attack, Sec5.1) and increases leakage (diverse models provide complementary views of the training data, Th5.6). These, as well as set stability (Th5.4-5.5), are set-level phenomena not visible when evaluating models individually. Also, Hsu et al focus on trees, while our theory covers trees, lists, and linear models, and our experiments extend to NNs (Fig18–20)
>
> **W2+Q2 Reactive robustness, its cost and policy object insufficiently described**
>
> We will revise to make this explicit. The intended scheme is:
> - One model from the Rashomon set is deployed, the rest is retained internally
> - The adversary targets only the deployed model as they have no access to the internal Rashomon set
> - When a vulnerability is detected via monitoring, auditing, or red-teaming, a replacement with high diversity (eg Hamming distance) from the compromised model on flagged inputs is selected
> - The replacement maintains near-optimal accuracy by construction
>
> Practically, reactive robustness forces adversaries to attack multiple models rather than one, increasing attack difficulty both computationally and theoretically. Adversaries also have fewer queries to work with
>
> No retraining is required, cost is linear search over the Rashomon set - cheaper than retraining a sparse optimal tree (NP-hard). The memory cost comes from storing the Rashomon set. For NNs, this cost is partly absorbed in practice as sometimes we already store multiple trained NNs as checkpoints from hyperparameter sweeps, random seeds or model search, approximating a Rashomon set implicitly. (This is what we mean by 'natural policy object': near-optimal models generated during development already exist in many pipelines.) For NNs, rotation means loading different weights with no retraining, which is practical for model complexity in our setting. Compact storage representations are available for sparse trees (Xin et al, 2022), where a Rashomon set is compressed (eg ~ 2MB for 180k trees), while a random forest stores each tree explicitly (eg ~2MB for 1k trees). Unlike random forests, we never aggregate predictions. The Rashomon set is held internally and a single interpretable tree is deployed at any time. Users always interact with one model
>
> Eq1 bounds transferability through prediction disagreement regardless of attack mechanism (Apruzzese et al 2023)
>
> **W3+Q3 Disconnected theory and how the tradeoff emerge**
>
> The unifying mechanism across the paper (Tab1) is *diversity*. Diversity within the Rashomon set blocks attack transferability (Sec5.1) and gives an adversary complementary views of the training data. Th5.6 quantifies the leakage rate, not just the direction, as a function of prediction variance across the set. This is exactly why the tradeoff arises: the same property that makes the Rashomon set useful for robustness also increases privacy risk. What changes across hypothesis spaces is how diversity is measured, which is inherent to the loss and model space: discrete models use combinatorial notions such as Hamming distance, while continuous models use geometric notions such as angular distance. We study multiple hypothesis spaces to show that this tradeoff is a consequence of diversity itself, not of one specific model class. Fig4,20 summarize this duality empirically for trees and NNs. We will add bridging sentences to make this connection more explicit
>
> We use KL divergence not to equate it with differential privacy (which is designed for one model), but to study *distributional leakage* under release of multiple models. In that setting, KL divergence and reconstruction error are the reasonable tools.
> Th5.4-5.5 are stability results, not equivalence results. The epsilon expansion is 2K/n, negligible for typical dataset sizes, confirmed empirically in Fig10
>
> **W1 Contributions feel anecdotic**
>
> The “Systemizing Multiplicity” survey mentions robustness and privacy as general desiderata, but does not analyze their interaction at the Rashomon-set level. To the best of our knowledge, ours is the first formal characterization of the robustness–leakage tradeoff at the Rashomon-set level, with both proofs and empirical validation
>
> **W4 Even sampling** selects models uniformly across the Rashomon set sorted by Hamming distance to the optimal model, ensuring the full range of diversity is represented, as opposed to sampling only from the boundaries

---

> > ### Author Rebuttal · Reviewer_xRwu · 2026-04-02
> >
> > Dear authors, thanks for the rebuttal that has clarified some points.
> >
> > *** Given the subtle difference with Hsu et al. (2025), including this positioning in the original submission would have imho been a better endeavour. In the actual related works (line 110 &sec) section, the most recent cited work is 4 years old.
> >
> > *** The reactive robustness proposed is sound, but eludes the principal difficulty in my opinion (detecting vulnerability). Using your detector, one could construct a simpler scheme for binary classification: i/deploy model ii/ when a vulnerability is detected, flip model decision. Should provide a fully robust model assuming a perfect detector -- and with a unique model.
> >
> > *** Regarding the theory results: sorry, perhaps my point was unclear. I meant that these theoretical results feel scattered because each introduces specific hypotheses. In the rebuttal you mention "our theory covers trees, lists, and linear models": I disagree with the wording: for instance, in thm4.2 you conclude single model is vulnerable .. for decision lists, and in 5.1 you establish diversity increases adversarial robustness .. for linear classifiers. So such a theory would only hold for models satisfying both 4.2 and 5.1 hypotheses, in other words models that are at the same time linear classifiers and decision trees. That sounds very rare..
> >
> > ***   Thank you for clarifying the role of diversity (which I believe should then be highlighted in the document, given a proper definition environment etc.), let us call it dd. To be clear I missed a "wrap up" that says: for a set of models MM of diversity dd, privacy=f(d)f(d) and robustness=g(d)g(d) for some f,g that would explicitely state this trade off. As a minor note, it would be great if dd was also a central metric of the experimental part, currently it feels mostly approximated by "number of trees", which is indirect. Especially for NN where uniform rashomon sampling does not feel granted.
> >
> > I'm happy to raise my score but stand on the negative side

---

> > > ### Author Response · Authors · 2026-04-05
> > >
> > > Thank you for engaging with us! Please see below:
> > >
> > > **W1: Related works**
> > >
> > > We have engaged with Hsu et al, 2025 thoroughly and will cite it accordingly. The related work section cites important papers in the subfields. It also cites several recent works from 2023–2025, including Boner et al, 2024; Meyer et al, 2024; Langlade et al, 2025; Dai et al, 2025; and Nguyen et al, 2025. We will add more citations for the highlighted section on line 110 [1,2,3] as well as other subsections. None of these works study robustness, privacy, and their trade-off at the Rashomon set level.
> > >
> > > [1] https://arxiv.org/abs/2306.08656
> > > [2] https://arxiv.org/abs/2302.04787
> > > [3] https://arxiv.org/abs/2412.11066
> > >
> > > **W2: Detecting vulnerability and post-hoc attack**
> > >
> > > While a sufficiently powerful detector can produce a fully robust model, in practice detectors often cannot identify which specific inputs have been attacked and can only detect attacks post-prediction. E.g., a spike in queries may indicate an attack, but it may not be possible to tell which specific inputs are adversarial, making it unreasonable to flip their predictions (which also decreases original accuracy). If queries require an immediate response, a prediction may need to be given before the unusual behavior is detected. Swapping to a diverse model in the Rashomon set can resist previously-fitted attacks while keeping high accuracy, without the need to know whether a given input is adversarial at inference time.
> > >
> > > As to if reasonable detectors can be built, there are already established practices for monitoring deployed ML/AI systems. For example, MLOps continuously monitors models in production to identify issues such as drift, degradation, and other anomalies (Kreuzberger et al, 2022). Tools such as Evidently and NannyML support this type of monitoring, including when ground-truth labels are unavailable. Evidently also supports stress-testing for adversarial behavior. With an internal Rashomon set, when an issue arises, a new near-optimal model could be swapped in immediately, without the need for retraining from scratch. This Rashomon-based process could potentially be automated as part of existing MLOps.
> > >
> > > Kreuzberger et al., 2022 - https://arxiv.org/pdf/2205.02302
> > >
> > > **W3: Theory across different HS**
> > >
> > > All of the arguments that are presented in the paper (single model is vulnerable, diversity helps robustness and increases leakage, Rashomon set is stable) apply to discrete models and to linear models separately. Those arguments should not be conflicted between hypothesis spaces. We present multiple diversity metrics and hypothesis spaces for generality of our proofs and empirical observations. Note that section 5.1 describes reactive robustness for 0-1 loss and *any* hypothesis space which includes discrete models (trees, lists).  We collected the details in the table (anonymized link https://ibb.co/4wm3yL0y) to help the reviewer navigate arguments between distinct hypothesis spaces.
> > >
> > > **W4: Trade-off parameterization**
> > >
> > > Thank you to the reviewer for clarifying the weakness. Below we take the core mechanism of the trade-off discussed in the paper and make it more mathematically explicit for 0-1 loss and discrete models with Hamming distance as a diversity metric.
> > >
> > > Within the paper notation, let $\bar d = \frac{1}{N^2} \sum_{i,j} H(f_i,f_j)$ be the diversity of the Rashomon set and $\bar d_{\hat f}= \frac{1}{N} \sum_{i=1}^N H(f_i,\hat f)$ be the diversity given $\hat f$, where H is Hamming distance.
> > >
> > > In the setting of Th5.6, we proved that $$var = \frac{1}{n} \sum_{k=1}^n \sigma^2(x_k) = 0.5 \bar d  \leq \bar d_f,$$ using the variance definition, algebraic calculation, and the triangle inequality.
> > >
> > > Through Th5.6, this allows us to bound the leakage by the diversity $\bar d_f$ of the Rashomon set. We can further show using Eq (1) that the adversarial accuracy under attack transfer from $\hat f$ is $$AvgAdvLoss = \frac{1}{N}\sum_{i=1}^NL_{S’}(f_i)\leq L_{S’}(\hat f)+\frac{1}{N} \sum_{i=1}^NH(f_i, \hat f)=L_{S’}(\hat f)+\bar d_{\hat f}$$
> > >
> > > **These two bounds make the tradeoff explicit: the same diversity ($\bar{d}_{\hat{f}}$) that can improve the robustness bound can also increase the leakage bound, providing a formal quantification of the double-edged nature of the Rashomon set for discrete hypothesis spaces**
> > >
> > > In our experiments (eg Figs 3,8), we do *not* equate the number of models with a diversity measure. The “number of trees'' x-label simply denotes the size of the sets of selected models, while Hamming distance is the explicit diversity selection criterion used to construct those sets (Sec 6.2-6.3, and App C.3). Our comparison of the “closest” and “farthest” strategies explicitly show a set of selected models with larger diversity are less private but more robust when compared to less diverse sets of the same size (Figs 8, 14; note that axes names have typo: x-axis is reconstruction error, y-axis is adv accuracy)
> > >
> > > We will add all discussed details to the paper

---

### Official Review · Reviewer_9HCq · 2026-03-13

**Soundness:** 3
**Presentation:** 4
**Significance:** 3
**Originality:** 4
**Overall Recommendation:** 5
**Confidence:** 3

**Summary:**

The paper investigates properties of the functions in the Rashomon set.  The Rashomon set consists of functions that are close to the ERM solution for a learning problem of interest.  In this direction the authors investigate robustness and privacy properties of the Rashomon set.  The main finding of the paper is that by having access to a diverse set of functions in the Rashomon set, this allows for two things to happen.

- First, the learner may defend against evasion attacks (adversarial examples) by not relying only on one function (the ERM solution) but rather by choosing a function from the Rashomon set in order to answer queries at test time.  The idea is that as the functions in the Rashomon set can be diverse enough, then they may have sufficiently different decision boundaries so that small perturbations of an initial input cannot cause misclassification.

- Second, when one uses several functions from the Rashomon set in order to answer queries, unfortunately this results to higher information leakage.  This is because each additional function exposed from the Rashomon set allows an attacker to have yet another "view" of the dataset.

Eventually the authors validate their theoretical findings using TreeFARMS that calculates the Rashomon set of sparse decision trees on six different datasets (Adult, Bank, COMPAS, Credit, Diabetes, FICO).

**Compliance With Llm Reviewing Policy:**

Affirmed.

**Final Justification:**

I am happy with the response that I received from the authors. I like the paper in general and I would like to recommend acceptance.

**Key Questions For Authors:**

**Q1.**  Do your conclusions on robustness extend to the situation where one would use the true label at the perturbed point rather than the label at the original point?  That is, instead of the definition $f(x_i') \neq y_i$ that you have in line 165, one would use $f(x_i') \neq y_i'$ such that $y_i'$ would correspond to the true label of $x_i'$?

**Q2.**  In the paper you mention (e.g., in line 200) rule lists as a plausible hypothesis space.  Do you refer to decision lists?  What kind of Boolean functions (or others?) are allowed in each node of such a decision list?  For example, a $k$-decision list has conjunctions involving up to $k$ literals in every decision node.  Can you please give an example of what you have in mind?  Especially, if you don't have $k$-decision lists in mind, perhaps give an example that is somehow different so that the distinction is clearer.

**Q3.**  While you study the properties of the Rashomon set in the paper it is a little bit unclear to me what strategy you suggest for prediction.  Should one select one of the functions from the Rashomon set at random and predict?  Should they use all of them and take the majority vote among them?  Should the majority vote be restricted on a subset of functions that are sufficiently diverse so that one can have some nice properties from your theorems?  Should something else be done?  Can you please clarify your stance on this.

**Limitations:**

yes

**Strengths And Weaknesses:**

The paper is well-written and has several interesting results.  Proofs are provided in the appendix, though I did not have the time to go through most of them, everything seems reasonable.  I think the paper investigates a reasonable problem that makes a lot of sense in the real world.

I have one small remark regarding adversarial examples.  Indeed, the way these are defined in the paper (lines 164-165) there are lines of work that indicate that there is an accuracy-robustness tradeoff.  However, ultimately this is unreasonable because, for example, if we know the ground truth function precisely, then such a model cannot err on perturbed inputs.  However, because of the definition in line 165, one may in fact charge an error (misclassification) to perturbed points for which the ground truth function at the perturbed point has changed.  For example, the starting point $x_i$ was very close to the decision boundary and now the perturbed point $x_i'$ is on the other side of the decision boundary.  In such a situation one in fact would prefer to predict the correct label (which is different than the original label) but the definition in line 165 would count this as a misclassification, when it is not.  There are lines of work that clarify these points and perhaps the authors could clarify the notion of robustness in the paper as well - assuming their results do not hold for other definitions of adversarial examples as well.  For example, in [R1] we have "prediction change" (not used by many people any more), "corrupted inputs" (definition in line 165 in this paper), and "error region" definitions for adversarial examples, depending on what is used for comparing the value of $f(x_i')$ against something else.  In [R2] one finds "constant in the ball" definition (which is identical to definition in line 165 in this paper) and "exact in the ball" definition which is identical to "error region" in [R1] and has to do with the true label at the perturbed point rather than the label at the original point.  Along these lines, when one uses the "error region" or "exact in the ball" definition for adversarial examples, robustness is no longer at odds with accuracy.

[R1] Dimitrios I. Diochnos, Saeed Mahloujifar, Mohammad Mahmoody. Adversarial Risk and Robustness: General Definitions and Implications for the Uniform Distribution. NeurIPS 2018: 10380-10389

[R2] Pascale Gourdeau, Varun Kanade, Marta Kwiatkowska, James Worrell. On the Hardness of Robust Classification. J. Mach. Learn. Res. 22: 273:1-273:29 (2021)

---

> ### Author Rebuttal · Authors · 2026-03-31
>
> Thank you for your thoughtful and encouraging review! Please see below our responses to the questions:
>
> **Q1+W1:  Do the robustness conclusions extend to using the true label at the perturbed point rather than the original label?**
>
> We appreciate the reviewer’s comments about constant-in-the-ball versus exact-in-the-ball robustness, and we will clarify in the paper that we are focusing on constant-in-the-ball robustness because, in practice, the true label at a perturbed point is not accessible.
> Regarding whether our conclusions extend qualitatively: our diversity result (Theorem 5.1, Corollary 5.2) extends directly to the exact-in-the-ball setting. These results depend on a geometric relationship (angle) between a model and the attacked model, which should hold independently of how adversarial examples are defined. For single-model vulnerability (Theorem 4.2), the exact-in-the-ball case is more nuanced. Sparse models with large prediction regions may in fact be robust when the true label is constant within those regions. Our formal bounds do not transfer directly to this setting, since Theorem 4.2 counts misclassifications relative to the original label $y_i$, whereas exact-in-the-ball robustness measures errors relative to the true label at the perturbed point $y_{\text{true}}(x')$. Points near the true decision boundary where the label genuinely changes would be counted as errors under our definition but not under exact-in-the-ball.
> For our empirical results, it is difficult to evaluate exact-in-the-ball robustness on real-world datasets since we do not have access to the underlying distribution. We think it may be interesting to investigate exact-in-the-ball robustness both theoretically and experimentally using synthetic data, though we leave this for future work.
>
>
> **Q2:  What exactly are rule lists?**
> Yes, to our knowledge rule lists and decision lists are equivalent and we use the terminology from Angelino et al 2018. In our analysis, each decision node contains a single literal (ie, a single binary feature), which corresponds to a 1-decision list. For example, a rule list in our setting might look like: *if employed $= 1$, predict positive; else if age $>50$, predict positive; else predict negative*.
> Our vulnerability result (Theorem 4.2) is derived for this 1-decision list setting. We believe the result can be generalized to $k$-decision lists by increasing the strength of the attack to $k$ (rather than 1), since an attacker would need to flip up to $k$ features to escape a conjunction of $k$ literals. Furthermore, if the attack strength is strictly less than $k$, one can construct a dataset and $k$-decision list such that no adversarial attack succeeds, but this requires data points to be far from the decision boundary. We will make this discussion explicit in the revision.
>
> **Q3:  What is the recommended prediction strategy when using the Rashomon set?**
>
> We thank the reviewer for this insightful question. We would suggest that for a specific (especially high-stakes) downstream task, select **one** model that aligns best with the domain knowledge or special constraints such as fairness. If no such criteria are available, random selection from the Rashomon set is a reasonable strategy, since all models in the set are near-optimal by definition.
> We do not recommend deploying the entire Rashomon set simultaneously as an ensemble. Using a large ensemble eliminates the interpretability inherent to individual sparse models and, as Theorem 5.6 demonstrates, exposes the system to increased privacy risks. While Appendix B shows that a sufficiently diverse ensemble can improve robustness by reducing attack transferability, this benefit comes at a privacy cost and sacrifices interpretability.
> In case of single model selection, the unselected models should be retained internally to provide reactive robustness. If the single deployed model is found to be problematic, the institution does not need to retrain a model from scratch. They can simply query their offline Rashomon set and swap in an alternative, near-optimal model that behaves correctly on the problematic points. Section 6.4 provides a framework for deciding how large a Rashomon set to retain internally, balancing robustness coverage against privacy risk.
> Formulating exact algorithmic strategies for model selection is highly task-dependent. We view the development of such diversity-aware selection as a direction for future studies.

---

> > ### Author Rebuttal · Reviewer_9HCq · 2026-04-04
> >
> > I want to thank the reviewers for their response.  I would like to maintain my score of suggesting acceptance (5).

---

### Decision · Program_Chairs · 2026-04-30

**Decision:**

Accept (spotlight)

**Comment:**

The paper provides an analysis of the two facets of the Rashomon set. On the one hand, it makes a compelling argument that the Rashomon set is a natural "policy object" that should be measured, governed, and explored. On the other hand, disclosing Rashomon sets poses privacy risks through unwanted inferences and information leakage. The authors frame this trade-off as one between robustness and privacy induced by model multiplicity.

The reviewers had a favorable view of the paper, highlighting that it is well-written and that it investigates a real-world problem (reviewers 9HCq, 2axa) and points toward an intriguing paradigm shift in which Rashomon sets are disclosed instead of individual models (xRwu).

Reviewer 2axa raised several questions about assumptions and formulations in the text, and noted that their concerns were adequately addressed by the authors in their rebuttal. Reviewer xRwu maintained a negative stance post-discussion and did not engage with the authors' last message addressing their concerns. My read of the rebuttal exchange is that the authors appropriately addressed the questions raised by xRwu.

The submission does an great job articulating the dichotomy between the risks and benefits of the Rashomon set. Though some of these insights may be "folklore" within the community of researchers studying this topic, the paper offers a precise and fresh take on the subject -- particularly the privacy angle -- and merits publication at ICML. I am sure this will lead to interesting discussions and follow-up work on model multiplicity.